# A framework for the emergence and analysis of language in social learning agents

Tobias J. Wieczorek[1,2], Tatjana Tchumatchenko [2], Carlos Wert-Carvajal[2,3] ✉ & Maximilian F. Eggl [2,3] ✉

Neural systems have evolved not only to solve environmental challenges through internal representations but also, under social constraints, to communicate these to conspecifics. In this work, we aim to understand the structure of these internal representations and how they may be optimized to transmit pertinent information from one individual to another. Thus, we build on previous teacher-student communication protocols to analyze the formation of individual and shared abstractions and their impact on task performance. We use reinforcement learning in grid-world mazes where a teacher network passes a message to a student to improve task performance. This framework allows us to relate environmental variables with individual and shared representations. We compress high-dimensional task information within a low-dimensional representational space to mimic natural language features. In coherence with previous results, we find that providing teacher information to the student leads to a higher task completion rate and an ability to generalize tasks it has not seen before. Further, optimizing message content to maximize student reward improves information encoding, suggesting that an accurate representation in the space of messages requires bi-directional input. These results highlight the role of language as a common representation among agents and its implications on generalization capabilities.

In exploring task representations in biological and artificial agents, studies have traditionally emphasized the role of self-experience and common circuitry priors[1,2]. Intriguingly, shared neural representations underlie similar behaviors among conspecifics[3]. Indeed, common convergent abstractions are also essential for communication among individuals of the same species or group[4]. Such social pressure implies that neural circuits may have evolved to produce internal representations that are not only useful for a given individual but also maximize communication efficacy, which has been argued to be essential in the development of cognition[5–7].

We posit that social communication is crucial in providing task-efficient representations underpinning the generalization of experiences among cooperative agents. The hypothesis that context and communication alter the task representation can be attributed to the introduction of language games[8] and supported by the ability of the neural activity to represent semantic hierarchies[9]. Early studies in this direction focused on the conditions and constraints that would allow an artificial language to evolve and how similar this construction would be to human communication[10–13]. With the advent of deep learning, there has been a surge of work that combines multi-agent systems with communication policies[14–20]. This includes studies on multi-agent games where agents send and receive discrete messages to perform tasks[21], translation tasks[22], and low-level policy formation through competition or collaboration[23]. Further work has highlighted the importance of pragmatic approaches[24], the contrast between scientific or applied models in language emergence[25], and multi-agent

[1]Department of Computer Science, Technical University Darmstadt, Darmstadt, Germany. [2]Institute of Experimental Epileptology and Cognition Research, University of Bonn Medical Center, Bonn, Germany. [3]These authors jointly supervised this work: Carlos Wert-Carvajal, Maximilian F. Eggl. ✉ e-mail: cwer1@uni-bonn.de; meggl@uni-bonn.de

cooperative learning[26]. However, these studies have focused more on the performance consequences of the communication system instead of examining the nature of shared representations.

To understand the interplay between the environmental experiences and the internal abstractions, we build on a teacher-student framework to develop a communication protocol that allows agents to cooperate while solving a common task[27,28]. We employ a reinforcement learning (RL) framework[29], which has been previously used in artificial agents[21,30], to produce empirical task abstractions that vary among agents. Using this RL-based student-teacher framework, we can recapitulate features considered to be critical for language[31], including "interchangeability"[32], where individuals can both send and receive messages[33], "total feedback", where speakers can hear and internally monitor their own speech[34] or "productivity", where individuals can create and understand an infinite number of messages that have not been expressed before[35].

In contrast with previous work, we focus on understanding how hidden representations can be shared between agents and what effect the structure of the lower-dimensional language space is. We are particularly interested in three aspects: how agents internally abstract real-world variables, how these abstractions translate into a common, shareable language, and the interaction of these elements. Hence, we opted for a non-discrete language model to directly compare the continuous nature of both brain processes and real-world phenomena. By feeding into the language model the learned information provided by RL agents performing a navigational task[36], we investigated the development of natural language as it arises from social and decision-making processes[37]. This leads to individualized abstractions emerging organically rather than being pre-defined, in contrast to supervised learning methods[22,38,39]. By analyzing the structure of the language embedding, we can gain insights into information content in the message space and its relation to neural representations underpinning task performance and generalization. Additionally, it stands apart from previous non-supervised, symbolic methods[28,40–43], taking cues from continuous language generation models[44,45] and animal communication systems, like the bee waggle dance, which translate a continuous environment into a concise message space[46,47], also seen in human languages[44,45].

To summarize, we present a tractable framework for studying emergent communication, drawing upon established multi-agent language models. We disentangle the relation between the internal neural representations and the message space, contributing to the following three results to the neuroscience and neuroAI communities:

- Reveal structural features in the lower-dimensional embedding space necessary for higher student success and task generalization[6,48].
- Demonstrate how the structure of the lower-dimensional embedding or message space is altered to enhance the information content when the communication channel is provided with feedback to optimize student performance.
- Understand how the hidden representations can be used in studying symmetric communication, i.e., where the sender and receiver can be interchanged, highlighted recently as an important challenge in the field[32].

## Results
### Model architecture
To study the emergence of language between agents, we define two agents passing information to each other: a teacher and a student. Both of these agents are modeled as deep neural networks, whereby the teacher network is trained in an RL framework, and the student learns to interpret the instructions of the teacher[29,49,50]. We used RL due to our interest in analyzing shared and generalizable abstractions arising from individual experiences and strategies instead of pre-determined labels. Additionally, RL provides an intuitive and robust

connection to neuroscience[29,51], which we aim to take advantage of to gain insight into the mechanisms and features of language emergence.

In our setup, the teacher agent is presented with a task with complete access to its observations and rewards (Fig. 1). After a certain amount of training, the teacher will have obtained sufficient information to represent the task. The teacher network aims to produce a state-action value function or Q-matrix ($Q(s, a)$) of the task, which contains the expected return of state-action pairs, hence learning in a model-free and off-policy form. The student then aims to solve the same task but with additional information from the teacher, which we will refer to as a "message" (Fig. 1a). Thus, the student must learn and complete the task through its observations and the message from the teacher. In our framework, we assume each teacher observes and learns from a single task and then passes a relevant task message—e.g., information derived from the Q-matrix—to the student. In that way, students can succeed on tasks they have yet to encounter by correctly interpreting the given information (Fig. 1b).

The most crucial component of the architecture is the communication process. Natural language represents a lower-dimensional representation of higher-dimensional concepts. When one individual speaks to another, high-dimensional descriptors - e.g., time, location, shape, context—of a concept in the brain of the sender are encoded into a low-dimensional vocabulary that is decoded back into a higher-dimensional and distributed representation in the brain of the receiver. This is supported by the observed semantic correlations and low-dimensional embedding space of human language representations[45,52,53] and in the brain activity[9,54,55], which is congruent across species[6,56]. To mimic this interaction, we introduced a sparse autoencoder (SAE)[57], that takes the information from the teacher and produces a compressed message, $m$, passed to the student alongside the task. SAEs are also neural networks that consist of two parts, an encoder, and a decoder, and promote sparsity for the lower-dimensional representations. The encoder continuously projects the teacher network's output, $Q$, onto a message, $m$, which is a real-valued vector of length $K$. The decoder then uses this message to minimize the difference between its reconstruction $Q'$ and the true $Q$.

Furthermore, inspired by the sparse coding hypothesis[54], we assume that the brain, and thus, by extension, language, is inherently sparsity-promoting[58,59]. We implemented this by adding the norm of the message vector to the autoencoder loss, which follows the principle of least effort to guide our artificial communication closer to natural language (see eq. (4) in the "Methods"). Here, we utilized the $L^1$-norm of the message vector, which leads to a promotion of zeroes in the message and, therefore, less information that mimics sparsity.

The combination of one teacher, SAE, and student for an arbitrary task can be seen in Fig. 1c, where we depict three different language-student interaction protocols. We note that this framework differs from an approach where all agents and languages are connected via one network. Instead, the teachers are always trained separately to generate the relevant task information. Then, we either sequentially train the language and student networks (no feedback in Fig. 1c) or connect the language and the student by providing the auto-encoder feedback on the student performance (Fig. 1c, with feedback). In essence, the teacher and the language (feedback and non-feedback) are connected conceptually through the information transfer process but not in a way that results in a single neural network or shared gradient flow. Variations of this approach were employed by Foerster et al.[34] and Tampuu et al.[60], which studied "independent Q-learning" where the agents each learn their own network parameters, treating the other agents as part of the environment. In this framework, we study a goal-directed navigational task in a grid-world maze (see "Methods", Fig. 1b). We chose this relatively simple toy problem for the agents to learn due to its straightforward implementation—allowing us to focus on analyzing the message structure—, its usefulness in studying generalization and exploration strategies, and the possibility

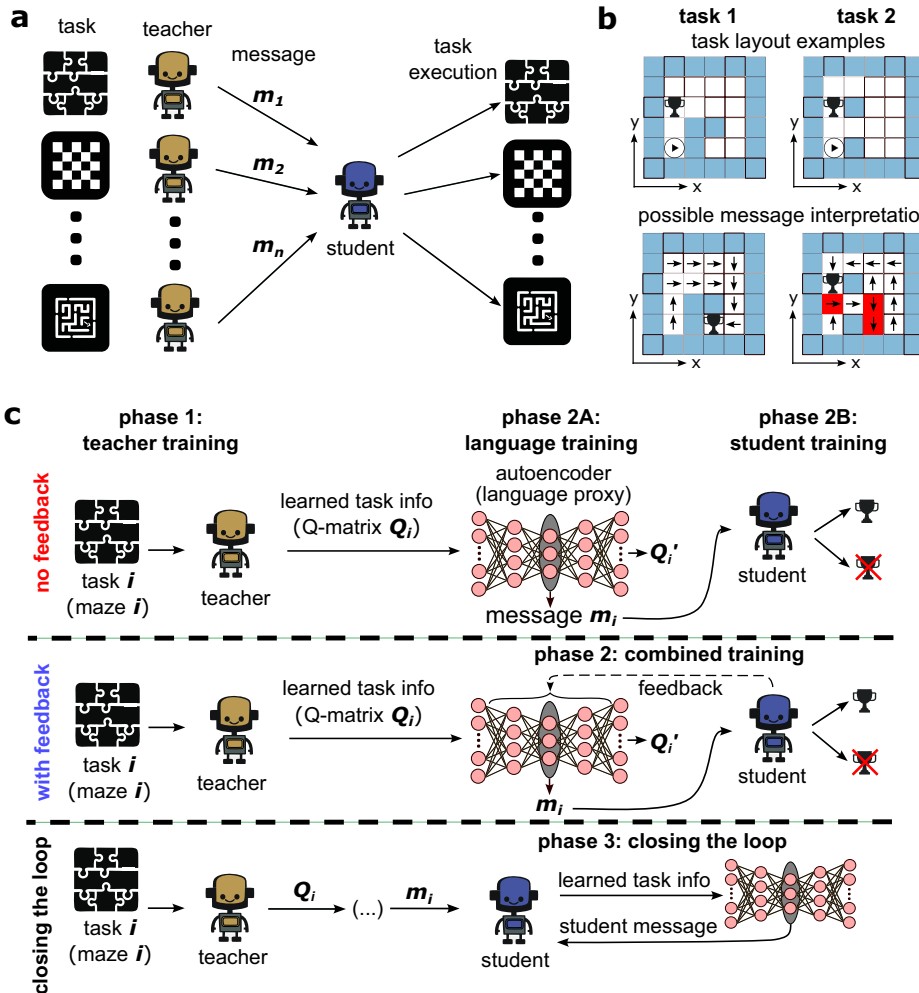

**Fig. 1 | Teacher-to-student communication model using a continuous compression of task solutions to low-dimensional message vectors. a** Model sketch depicting a generalist student agent that is provided messages from teacher agents for various tasks. The student learns to decode these messages and then perform the relevant tasks. **b** (Top) Representative navigation tasks used to train and test agents to analyze the social learning framework. Beginning in the bottom left corner, the agents aim to reach the goal (trophy) in as few steps as possible while avoiding the walls (light blue squares). (Bottom) Overlaid are example policies for tasks learned by the teacher agents. The student needs to decode the encoded version of this information it receives. Messages ($m_i$) may contain erroneous instructions or be misunderstood by the student (red squares). **c** Detailed communication architectures used in this study. In each of the three approaches, task information (Q-matrices in our framework) is first learned by teacher agents, who then pass this information through a sparse autoencoder (language proxy), which generates the associated low-dimensional representations, $m_i$. When student feedback is absent (top row), these representations $m_i$ are provided directly to the student who learns to interpret them to solve task $i$. In the case of student feedback (middle row), we also allow feedback from the student performance to propagate back to the language training and enhance the usefulness of the messages. The final schematic (bottom row) depicts the "closing-the-loop" architecture. Here, the student is trained on a set of messages from expert teachers. Once it is sufficiently competent, its task information is supplied to itself (after being passed through the language embedding trained with feedback), and the effect on performance is studied.

of extending the framework to more complex navigational settings[29]. We emphasize that the above architecture does not rely on a predetermined vocabulary for which the agents must assign meaning. Instead, the language evolves naturally from the task and the lower-dimensional encoding, mirroring natural language evolution.

The purpose of this study is two-fold: (i) analyze the structure of the lower-dimensional representations generated by the trained language (which are lower-dimensional representations of our tasks), and (ii) evaluate the performance of an agent who has learned to interpret a message coming from this embedding space.

## The structure of the lower-dimensional message

We trained a set of teachers to solve one maze task each with a specific goal location and wall setting. As mentioned above, we use the trained language to embed the Q-matrices into a lower-dimensional space—firstly, considering a language created without feedback by the

student. The resulting latent space shows wall positions as the most prominent dimension in the lower-dimensional representations (Fig. 2a(ii)), with goal locations being a secondary feature of the variability (Fig. 2a (iii)). This structure is represented in the lower-dimensional PCA through discrete groupings with minimal overlap and stratification within each grouping according to the goal location. This result follows intuitively from the fact that the language is trained without student feedback, only relying on the reconstruction of the Q-matrix and regularization of the message space (eq. (4)). Thus, to achieve this reconstruction most sparsely, a hierarchical structure appears: first, we distinguish mazes, and then, within each maze, we distinguish the goal location. This structure appears regardless of whether this information is helpful for the student. We note that when we used linear activations or singular value decomposition for the language encoding, we did not reproduce this clear grouping (cf. Supplementary Figs. S1 and S2).

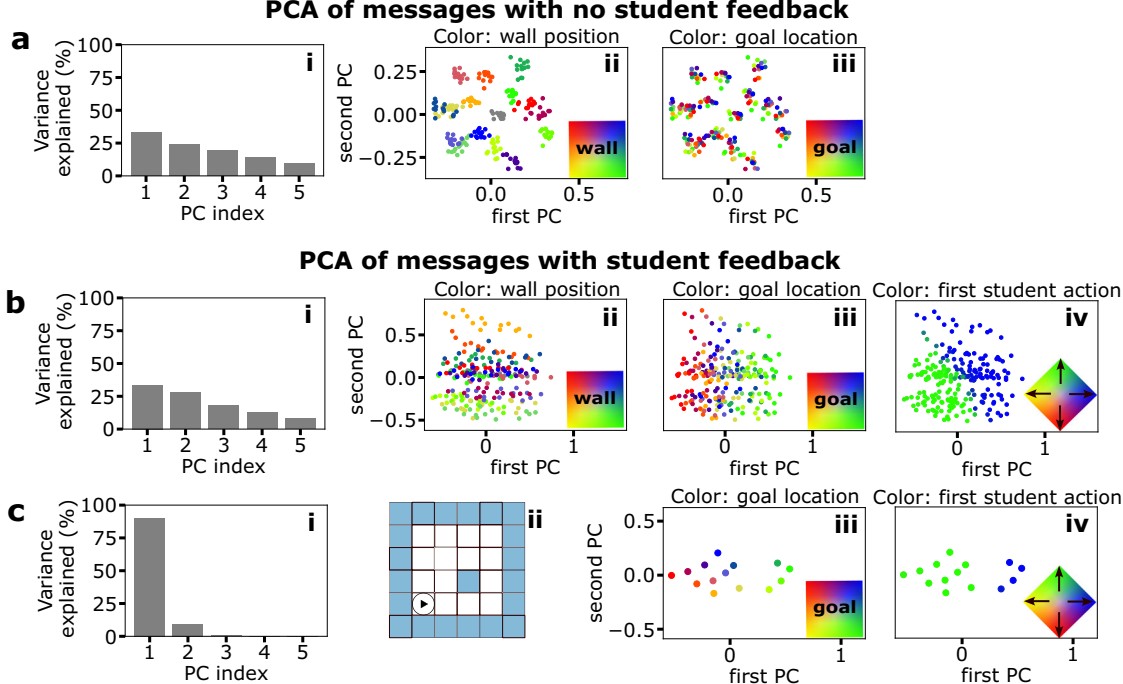

**Fig. 2 | Student feedback alters the language embedding according to a utility function. a** Principal Component Analysis (PCA) of the lower-dimensional messages of size $K = 5$ obtained from a language encoding without student feedback (eq. (4)) for all possible tasks in the $4 \times 4$ mazes with ≤1 walls (see "Methods" for a description of the tasks). i) Explained variance by principal component. ii)-iii) depicts the messages highlighted by the position of the single wall (gray refers to the maze with no walls) and by the position of the goal, respectively. **b** Result of the message encoding now including student feedback achieved by using eq. (1) for the loss function. i)-iii) depict the same concepts as in (**a**). iv) shows messages highlighted by the preferred first student action (step up or right). **c** PCA of the messages with student feedback from an example grid-world (depicted in ii)).

While direct labeling of the tasks by such dimensions may help the student solve trained tasks, the average performance concerning trained tasks and generalization is significantly lower than when student feedback helps shape the language (see Supplementary Figs. S3 and S4). Furthermore, this interaction is purely one-directional and does not reflect the natural emergence of language, which is a back-and-forth between the receiver and sender. Therefore, we introduced student feedback into the message structure to encourage this natural evolution of language. Such feedback is implemented by including and maximizing the probability of the student finding the goal in the language training. This translates to a compound autoencoder loss function of the form

$$\mathcal{L}_{\text{SAE, feedback}} = \mathcal{L}_{\text{SAE}} + \zeta \mathcal{L}_{\text{goal finding}} = \mathcal{L}_{\text{reconstruction}} + \mathcal{L}_{\text{sparsity}} + \zeta \mathcal{L}_{\text{goal finding}},$$
(1)

where the $\mathcal{L}_{\text{goal finding}}$ is defined by eq. (5) in the "Methods" and $\zeta$ is a tunable hyperparameter. After each trial of the student, the language is updated to (i) maintain the reconstruction of the information, (ii) promote sparsity of the message, and (iii) increase the success rate of the student given the message.

Notably, the latent structure of the language space significantly changes through this reward-maximizing term (Fig. 2b(ii)-(iv)). Even if the variance distribution remains similar (compare Fig. 2a(i),b(i)), task settings are no longer clustered in the latent space, but instead form a more continuous gradient when marked by wall position (Fig. 2b(ii)) or goal location (Fig. 2b(iii)). Therefore, the feedback changes the lower-dimensional task representations so that the student obtains more information on where to go, i.e., the policy, rather than the actual composition of the state space. We note some overlap in the middle of the cluster when marking the tasks by goal location; here, the policy differences are negligible as there might be two competing policies that are equally optimal. This focus on policy is additionally emphasized by the variability along the initial action of the student (Fig. 2b(iv)), where a clear split between the two choices of going right or up can be observed. By providing this policy label, language moves away from providing maze labels and towards a framework that can generalize to tasks the student has not seen before. Table 1 shows the changes in explained variability by wall position and goal location in both languages without and with student feedback. Notably, the message variability between groups of goal locations (see Methods) rises when the utility constraint is introduced, marking the increased importance of describing the goal location accurately in the language.

This focus on policy, rather than state space, appears to be independent of the architecture of the autoencoder we use (cf. all linear activations in Supplementary Fig. S1) or the dimensionality reduction technique we employ (see Supplementary Figs. S5–S9 for results using PCA, UMAP, and t-SNE, related within and between variances in Supplementary Table S1). Additionally, the projection of messages to the main dimensions of a linear decoder was consistent with the unsupervised representational space (Supplementary Fig. S10). This implies that transmitting this representational feature is fundamental to the success of the student.

We can extend this analysis to understand the student feedback representation of the different goal locations for a single maze, where more than 80% of the variance is explained by a single principal component (Fig. 2c(i)). Geometric structure (Fig. 2c(iii)) and action selectivity (Fig. 2c(iv)) are well represented in the embedding, the former indicating that language is performing a simple linear transformation of the geometric shape of the maze. We hypothesize that such information hierarchy benefits overall learning and generalization. We note that these results hold independent of the activation function (Supplementary Figs. S1, S2).

**Table 1 | Analysis of variance for world groups and goal groups in the message spaces from Fig. 2**

| Message grouping | Var$_{within}$(X) | Var$_{between}$(X) | β | F-value |
|---|---|---|---|---|
| By wall position (Fig. 2a(ii)) | 2.88 | 20.18 | 0.875 | 97.54[a] |
| By goal location (Fig. 2a(iii)) | 19.99 | 3.07 | 0.133 | 2.14[a] |
| By wall position with student feedback (Fig. 2b(ii)) | 20.06 | 38.07 | 0.655 | 26.44[a] |
| By goal location with student feedback (Fig. 2b(iii)) | 38.26 | 19.87 | 0.342 | 7.24[a] |

The statistical test performed was a one-sided F-test for variance analysis of groups (for more details see the statistical analysis described in the "Methods").
[a]Refers to a significant difference in group means with significance level set at $p = 0.05$.

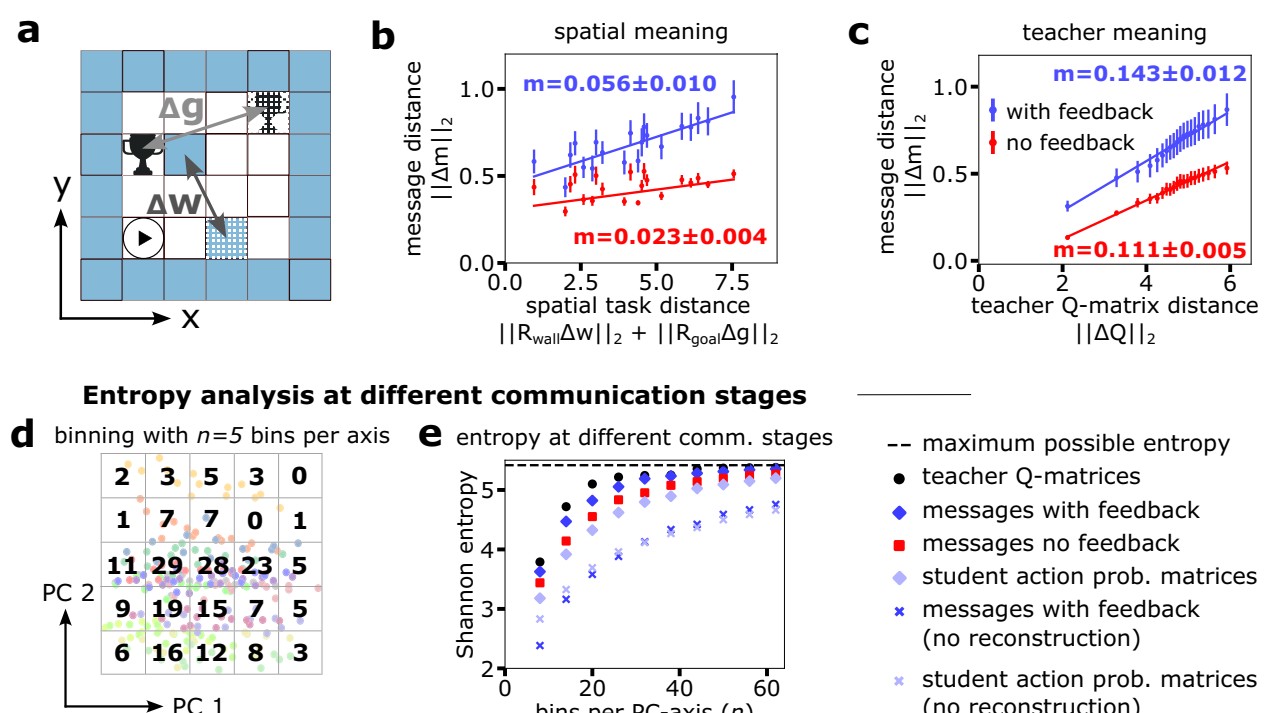

**Fig. 3 | Topographic similarity and entropy analysis of emergent languages.**
**a** Visualization of goal ($\Delta g$) and wall ($\Delta w$) distance vectors between two tasks (task 1: solid, task 2: checkered). In combination, these are used to compute a spatial task distance $\Delta t = ||R_{wall}\Delta w||_2 + ||R_{goal}\Delta g||_2$ for (**b**). **b**, **c** Comparison of pairwise task meaning distances and pairwise distances of the corresponding messages. Message and task distances are measured with the Euclidean norm, Q-matrix distances with the equivalent for matrices, the Frobenius norm. Center and end points of the bars refer to the mean over five languages and ± one standard deviation, respectively. *m*

refers to the gradient of linear fit. **d** Entropy analysis through discretization: The first two PCs of normalized samples of each data type are binned. The entropy is then computed for different choices of bin size. Here, we depict an example discretization for 5 bins in each PC direction (bin side lengths are identical along both axes). **e** Calculating the entropy for each PC discretization demonstrates a clear ranking of the teacher, message, and student information-carrying capabilities. The maximum possible entropy is that of a uniform distribution over all maze tasks.

One crucial feature of language is its compositionality, which neighborhood properties can measure; for example, close meanings in a compositional language should map to nearby messages. To test whether our communication protocol includes this feature in its mapping from meaning to message space, we performed topographic similarity analysis[21,61] by comparing the distances in the message space (Euclidean distance) against (i) the task labels, i.e., spatial difference in the mazes and (ii) the information that the teacher provides to the autoencoder (Q-matrices). The distance in the labels was calculated as a weighted sum of the differences between the goal and the wall locations (see Fig. 3a). In contrast, the distance of the teacher Q-matrices, representing a space-based and action-based meaning, was calculated using the Frobenius norm.

The calculated pairwise distances are plotted in Fig. 3b, c. The message and meaning spaces show topographic similarity as indicated by the positive slope parameters of the linear regressions. Taking this slope as a quantitative measure, we also find that languages trained

with feedback show higher degrees of topographic similarity (and thus compositionality) than languages trained without feedback (cf. blue vs red).

Another property of emergent (discrete) communication is its tendency to minimize entropy[62], i.e., the general pressure towards communication efficiency. To see whether this effect was reproduced in our framework, we performed an information theoretic analysis using Shannon entropy to measure the information-carrying capabilities of our messages. As Shannon entropy is restricted to random variables taking discrete values and our messages arise from a continuous embedding space, we projected our messages onto a set of bins and then calculated the entropy of that discrete distribution. For visualization purposes, we only show the first two PCs and normalized the distributions with an identical factor along both PC axes so that samples were restricted to $[-1, 1]^2$ (see Fig. 3d for an example). While this means that some information is omitted, direct comparison of the entropy of tasks (the maximum entropy value), teacher outputs,

messages, and student output becomes possible. We calculated the entropy across various bin widths to verify that our findings are independent of our chosen bin size.

We found that the entropy decreases when moving through the communication framework, i.e., the entropy of the teacher outputs is highest, followed by the entropy of the messages and finally the entropy of the student outputs (see Fig. 3e). This result is intuitive because, at each stage, information is lost as it passes through an agent/network. Nonetheless, when comparing the framework where the auto-encoder is provided with feedback on the student performance to the one without feedback, we see that more information is retained. This implies that bi-directionality is critical to the ability of the language to transmit helpful information.

Additionally, we confirm one of the findings of ref. 62, which states that there is pressure for a language to be as simple as possible and that this pressure is amplified as we increase communication channel discreteness. We simulate this scenario by removing the reconstruction loss from the auto-encoder training so that the auto-encoder loss only consists of the sparsity promotion and the student performance feedback. This amplifies the pressure of the auto-encoder to generate a sparse message space. We find that the entropy of the messages and student output is significantly lower than the situation with the reconstruction loss (no reconstruction data in Fig. 3e). Furthermore, the entropy values of the messages and student output do not differ significantly, meaning that the messages and student output have collapsed onto distributions with similar information-carrying capabilities.

Interestingly, the addition of student feedback reduced the overall reconstruction error of the message space (Fig. 4a–c, Eq. (4)). This may suggest that the reconstruction of the teacher Q-matrix benefits from including features guided by utility and transmissibility criteria. Nevertheless, this comes at the cost of lower sparsity (Fig. 4c), mirroring the effect of natural language: communication aims to transmit the most sparse message, allowing for the best reconstruction of the underlying idea. Overall, these three items achieve a similar level of compound loss in both feedback and non-feedback (Fig. 4d).

### The effect of the message on student performance

In order to test the performance and generalization capabilities of the student, we used messages from teachers who mastered mazes with zero or one wall state and trained the student on patterned subsets of their goal locations (Fig. 4e, inset). We define the task solve rate as the percentage of goals attained under $2s_{opt}$ steps, where $s_{opt}$ corresponds to the shortest path from start to goal (see "Methods").

Under these terms, we can observe an increased performance of the student against misinformed students (given incorrect messages) and random walkers (one of which avoids walls) when evaluating the trained goal sets (Fig. 4e). We note that even in this scenario, this misinformed student slightly outperforms the random walkers, which we hypothesize is because of the initial action preference we observe for all messages, which allows the misinformed student to avoid the outer walls. To ascertain whether the generalization of the goal locations across the messages was achieved, we tested the performance of the students on unknown goals. We observe that the best generalization is achieved under checkerboard patterns. However, the performances of the other four cases do not differ significantly from the random walkers (Fig. 4f). This implies that generalization is difficult when large portions of the task space are unknown, and interpolation between known tasks is not possible. Training on far-away goal locations leads to slightly better performance (Fig. 4f(v), (vii)), but this might also be due to a wall-avoiding action preference. In this respect, when adding new wall locations, the overall performance is reduced, but the improvement against the other agents is preserved (Fig. 4g, h). These results highlight the importance of the goal-oriented structure of the lower-dimensional representations for these tasks and reinforce

the benefits of the altered language achieved by the student feedback. In line with previous observations (Fig. 2b), the main features of the encoded message are the policy and goal location. Therefore, when the agent attempts to solve the maze with unknown goals, it performs markedly worse. This behavior is only avoided when the student is trained on the checkerboard pattern, which means it has seen the entire maze and can use the information presented and its own experience to compensate for the lack of information. In other words, new tasks must be composable from other tasks within the language framework for communication to succeed.

We note that the above results arise from a language generated with student feedback, i.e., the representations that help the student have direct knowledge of the student parameters. To ascertain whether this language is useful to students who were not directly involved in the language training, we studied the performance of novel students who were trained to interpret "frozen" languages without feedback (Supplementary Fig. S3) and with feedback (Supplementary Fig. S4). We note that the former approach treats all the components of our framework (teacher, language, student) as separate networks and that no gradient information is propagated back through the language channel. We see that in both cases, the students perform well on many tasks and can generalize to unknown scenarios. However, the student trained to interpret the frozen feedback language performs better across all scenarios (and outperforms the smart walker). This implies that initial feedback is fundamental to generating helpful language features that other new students can use.

### Closing the loop

As natural language is not usually restricted to sender and receiver but is a robust exchange between two agents (i.e., "interchangeability"[31]), our final analysis is related to studying the effect of passing task information gained by the student through language encoding to obtain a set of novel messages. Rather than solely relying on a set of teachers that perform single tasks and pass on compressed information, we allow the student to generate messages itself after performing - and thus learning—tasks with messages from teachers. These student messages are then passed back to themselves, and their performance with these messages is assessed. A schematic depicting this structure can be seen in Fig. 1c (bottom row). Thus, we attempt to create a simple generalist agent to supply information through the same language encoding, which we keep fixed. This communication process will naturally erode the message, leading to comparisons to the children's game "telephone". What information is robust to communication erosion is often studied in that setting. We can use this analogy also to identify the type of information that is more transmissible between agents[63].

Firstly, we can observe a degradation of the information content. Notably, the low-dimensional form of the student-generated task information entails that variability among student messages is mainly concentrated on a single dimension that is identifiable by the goal location and initial action (Fig. 5a(i), Table 2). This contrasts with previous findings that the message space of teachers was not dominated by one principal component, and variability also corresponded to wall arrangements (Fig. 2b). We then turn to the task completion rates of the students. Here, both the informed and misinformed students are given messages resulting from encoding the task information the student has learned when supplied the teacher with messages. The informed student is supplied with the encoded message corresponding to the current task, while the misinformed student is provided a message from a random task. From a performance perspective, we note that the degradation of the message content translates into lower task solve rates (Fig. 5b–e). This decrease can be seen even when considering trained goal locations (Fig. 5b). Nevertheless, students performed better than the misinformed agents, which implies that

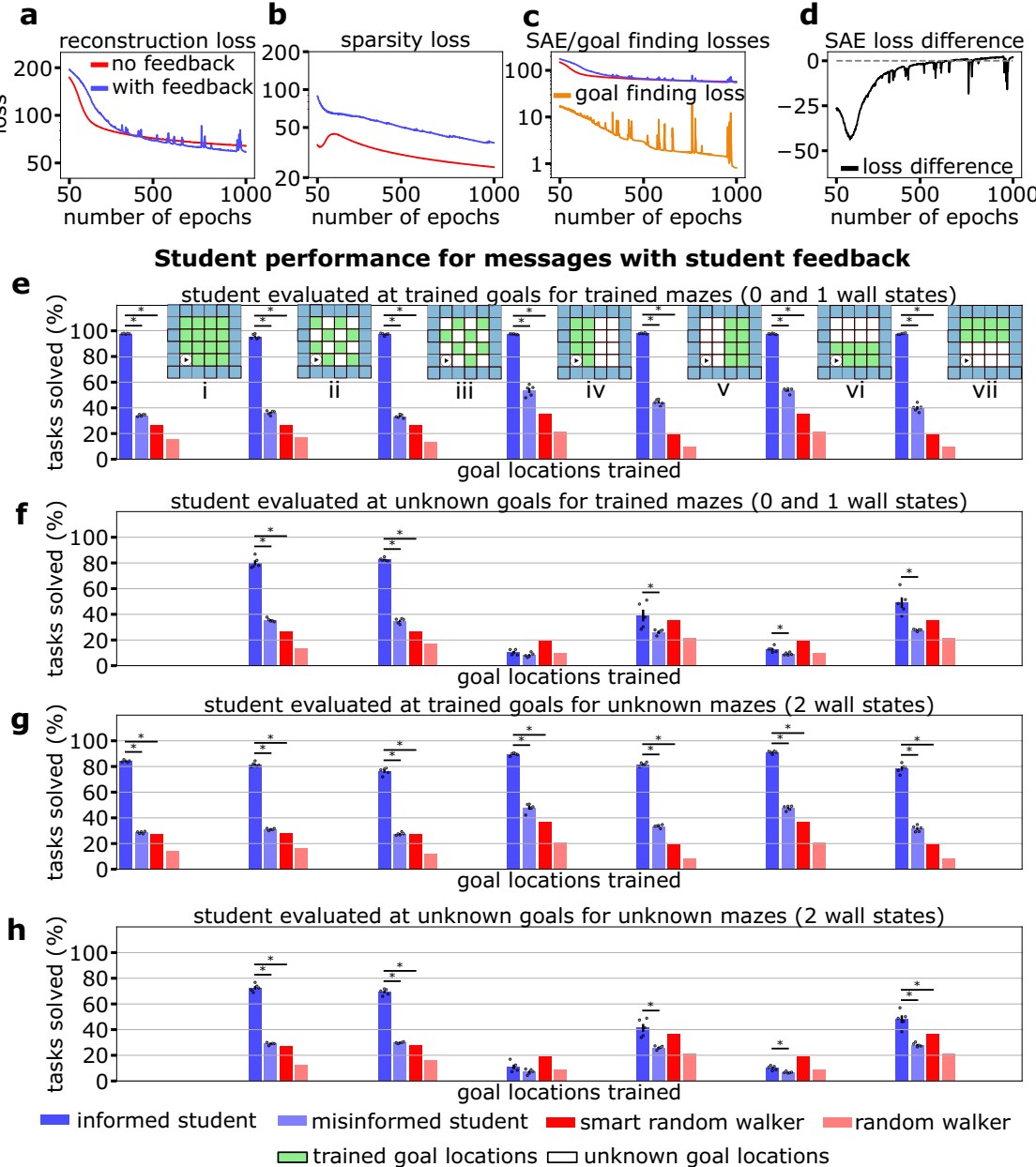

**Fig. 4 | Reward-based student feedback enhances performance, generalizability, and autoencoder reconstruction.** Components of the compound autoencoder training losses with and without student feedback; (**a**) reconstruction loss, (**b**) sparsity loss, and (**c**) SAE loss, which additionally includes the goal finding loss when student feedback is included (see Eq. (4) and Eq. (5)). **d** shows the difference $\mathcal{L}_{SAE} - \mathcal{L}_{SAE,feedback}$ which highlights that the student-feedback autoencoder achieves a lower reconstruction loss. **e**–**h** Student performance on training and test maze tasks (see "Methods" for a description of the tasks). A comparison is made between the informed student, who receives the correct message, the misinformed student, who receives a message corresponding to a random task, and two random walkers, one of which never walks into walls (smart random walker). The performance is further evaluated for seven sets of trained goal locations, i)-vii), displayed as the green squares in the inset figures of (**e**). In the 4 panels, the bars represent the mean task performance ± the SEM. In (**e**), the performance is measured for the trained goal locations and trained mazes with 0 or 1 wall state. **f** shows the solve rate for the unknown goal locations (white) for mazes with 0 or 1 wall state. **g**, **h** depict the solve rates for new mazes (2 wall states) with trained and unknown goal locations, respectively. The error bars for each case refer to the variance across languages (five languages were trained for each case). *refers to $p < 0.05$ where the $p$ is obtained using a two-sided and one-sided $t$-test (multiple tests were accounted for using a Bonferroni correction factor) for the informed vs misinformed and smart random walker, respectively. Exact $p$-values for the significant results can be found in Table 6.

passed degraded messages include sufficient information to avoid walls and find the goal state.

Given that the key features of the message arising from the lower-dimensional representations are the goal location and initial action features, it is unsurprising that, as long as the goals are known, the informed student performs well on maze tasks it has not seen before (Fig. 5d). When considering the performance of the student on unknown goals, for both trained (Fig. 5c) and untrained goal locations (Fig. 5e), we

note that, in most cases, the informed student performs, at most, on par with the smart random walker. This indicates that the message has a detrimental effect on the students. It can not generalize to the goals it has not seen, as the information provided does not allow it to build an adequate representation of the task. Finally, even though the degraded messages do not carry significant information about the world configuration, we hypothesize it is sufficient to produce minimally better performance in the known worlds compared to the unknown worlds.

**Fig. 5 | Encoding the student's task information, instead of the teacher's, and passing it to the student leads to lower task performance. Nonetheless, some task-relevant information benefiting the student remains. a** PCA on the student messages (encoded messages arising from the student task information): i) shows the variance explained by PC. ii)-iv) depict the student messages marked by wall position, goal location, and probability of initial student action, respectively. **b**–**e** Informed student performance on training and test maze tasks (see "Methods" for details) is compared against the misinformed student and two random walkers.

The comparison is once more performed for the seven sets of trained goal locations, (i)−(vii). In the four panels, the bars represent the mean task performance ± the SEM. The relevant tasks per panel are identical to Fig. 4. For (**b**−**e**), 25 languages were originally trained and evaluated, but a subset was excluded (see "Methods" for details on this exclusion). *refers to $p < 0.05$ where the $p$ is obtained using a two-sided and one-sided $t$-test (multiple tests were accounted for using a Bonferroni correction factor) for the informed vs misinformed and smart random walker, respectively. Exact $p$-values for the significant results can be found in Table 6.

We conclude that the student output retains pertinent task information that can enhance the performances of other students, even if degraded. This can be seen in the solve rates of the informed student. They are always higher than those of the misinformed student, allowing us to assume that the student can use relevant information within the degraded message. However, generalizability to unknown goals is lost under this framework, even when the student previously achieved high success rates (Fig. 4f, h, checkerboard).

Nevertheless, these results represent an early attempt to analyze task-driven communication with generalist agents. Notably, one key aspect is how a compromise or balance between tutoring and learning

can be achieved in multi-task and multi-agent systems to keep a relevant and generalizable message space. In other words, relevant features across tasks can be captured by a centralized embedding generated by individual experiences of agents, similar to how biological agents behave.

## Discussion

Task-relevant representations, either in the brain[2,64], as part of a linguistic system[4,65] or in artificial agents[1], ought to be generalizable. Humans take advantage of this generalizability to perform new or slightly different tasks from the ones they may have encountered

**Table 2 | Analysis of variance for world groups and goal groups in the message spaces from Fig. 5a**

| Message grouping | $\text{Var}_{within}(X)$ | $\text{Var}_{between}(X)$ | $\beta$ | $F$-value |
|---|---|---|---|---|
| By wall position (Fig. 5a(ii)) | 1583 | 54.5 | 0.033 | 0.48 |
| By goal location (Fig. 5a(iii)) | 367 | 1270 | 0.776 | 48.15[a] |

The statistical test performed was an one-sided $F$-test for variance analysis of groups (for more details see the statistical analysis described in the "Methods").
[a]Refers to a significant difference in group means with a significance level set at $p = 0.05$.

before. For instance, when learning to ride a bicycle, an individual does not need to relearn all the principles of balance and coordination when switching to a different bike or even another mode of transportation, like a scooter or a motorcycle. Similarly, an artificial agent faced with an out-of-distribution task may need to draw on its internal representations and their generalizability to complete it successfully. However, it remains an open question how social agents can reconcile abstractions from their own experience with those acquired through communication.

We present a multi-agent RL system of teacher-to-student communication that accounts for task-wide variability. Notably, embedding the state-action value function into a low-dimensional format leads to effective abstractions, enabling agents to learn goals and states flexibly across trials from model-free instructors. Additionally, we introduce a framework to analyze the nature of such communication protocols. Drawing inspiration from Tucker et al.[28], our research builds upon their findings that agents can effectively communicate in noisy environments by clustering tokens within a learned, continuous space. Additionally, we reference the work of Foerster et al.[34], who developed a model for independent parameter learning by agents, along with a system facilitating real-valued messages during centralized learning. Unlike the approach of Foerster et al., which shares gradient information between agents for end-to-end training, our method uses a continuous channel solely for task representations and trains agent parameters separately, without shared gradient data. This approach yielded a latent structure that prioritized variability along the goal space instead of the maze configuration, contrasting with the prominence of the state space in solely teacher-based models.

Using this framework, we studied the representational nature of the message space that includes the student return in training. We found this approach improved performance and yielded a latent structure that prioritized variability along the goal space instead of the maze configuration, in contrast with the prominence of the state space in the solely teacher-based one. Thus, by including reward-based constraints in language development, we saw that the communication channel could prioritize answering to task structure while acquiring a similar—or superior—reconstruction error. Akin to "total feedback"[31] in human speech, where the speaker modifies their message based on environmental factors/presence of other individuals, the student-autoencoder structure adjusted the message space based on the student performance.

Additionally, we studied the importance and sensitivity of this language space by feeding back the space-action value maps of students in a similar manner to the telephone game[63]. The degraded information confirms the relationship between the quality of representations of agents and their performance and points to the importance of good sample space to construct language. Despite this degradation, we retained certain features that were important for task performance (e.g., the ability to interpret novel messages using known messages—akin to "productivity"[31]), a similar effect that has been observed in human speech[66].

Overall, our results indicate that a generalist agent should be able to relate to the language space in an invariant manner. This opens up avenues to study the importance of specific social structures, such as teacher or student roles, which may be critical to a robust language space and to moderate the information flow.

The implications of our study suggest possible analogies with natural languages. First, our system evolves according to a utility or gain function, not solely to error minimization or comprehensibility. Lossless information transmission is insufficient for competent behavior, and the message space needs to adapt to be advantageous for other agents. This is similar to the natural language, where morphemes evolve according to motives, goals, and efficiency of a group[37,67]. For instance, in birds, it has been observed that utility drives the emergence of new linguistic relations or compositions[68,69]. Second, introducing dimensionality and sparsity constraints is motivated by anatomical and cognitive limitations, such as vocal tract size or memory capacity[70,71]. Hence, by allocating a predefined number of dimensions to our communication system, we replicate such properties and observe that these are organized into hierarchical task-relevant modes. However, ongoing work still aims to answer how channel size relates to the representational space, as machine learning and brain activity tend to converge to a high-dimensional space in the representations that are not shared by the actual symbolic space[55,72]. Studies have shown that brain activity is compressed relative to the message space even if our languages are not precisely low-dimensional[9,53].

In this study, we did not utilize sequential composition to generalize our message[39,73]. Instead, we aimed to generalize through interpolation of the continuous messages. Nonetheless, the framework can readily be extended to include composable messages using sequential sets of tasks (termed "duality of patterning" in the work of Hockett[31]), which will be the focus of future studies. Additionally, in contrast to natural language, our model lacks predefined syntax or grammar with respect to behavioral variables[74,75]. By encoding task-relevant information (e.g., Q-matrices), the lower-dimensional embedding space was biased to include features of the task information indirectly. This was observed in the emergent hierarchical latent structure obeying task variables and is similar to social species that show cultural or experience-dependent complexity in their linguistic traits, like non-primate mammals such as bottlenose dolphins[76] or naked-mole rats[77]. In this sense, we presume that the neural representations and circuitry of the agents evolve and rewire to enable social learning[5]. By doing so, we look at the interplay between the community scale and the cognitive one instead of fixing communication or neuronal representations. Thus, research around generalist agents performing dual roles as teachers and students is crucial. This involves creating agents with distinct sender and receiver units and an experience-based policy. Additionally, examining the impact of the social graph on language construction and expanding to further tasks is vital.

Thus, inspired by Tieleman et al.[48], who employed an encoder-decoder model to examine how community size influences message representations, future work should systematically examine diverse channel structures. This would permit, for instance, to reverse the student-teacher roles and understand how information emerges and propagates through embeddings. Additionally, it would be interesting to examine more model-agnostic outcomes using sequential network architectures, e.g., recurrent neural networks or transformers. Furthermore, since discrete communication protocols are more commonly used than the continuous approach in our work[28,40–43], they should be integrated into our existing framework. Finally, as

introduced by Dupoux[78], there are several features critical for the study of language emergence and language learning: (i) being computationally tractable, (ii) using realistic tasks that can be performed by real biological agents and (iii) use the results of those biological agents as benchmarks for the artificial agent performance. In this sense, while tractability is a key component of our framework, we emphasize its utility to neuroethologists, who can work with biological data within our framework to study brain activity in relation to language abilities in future studies.

In this sense, while tractability is a key component of our framework, we emphasize its utility to neuroethologists, who can readily harness our framework study brain activity in relation with language abilities.

In conclusion, we have introduced a multidisciplinary approach to studying language emergence using reinforcement agents and an encoding network. Instead of treating our system in a fashion akin to linguistics, we approach the communication problem as a top-down representation problem starting at the neural representations. This framework opens compelling avenues to generate hypotheses about the interplay between individual and collective behavior and those abstractions, both internal and external, emerging from social communication.

## Methods
### Teacher agent Q-learning
In our communication model, shown in Fig. 1, the navigation task solutions are learned by the teacher agent (implemented via a multilayer perceptron) via deep Q-learning[29]. The $Q$-value for action $a$ and state $s$, $Q(s, a)$, represents the agent's future maximum return achievable by any policy. Despite the small state-action space, using an artificial neural network provides the flexibility to apply the framework to future tasks that may have much larger state-action spaces. Concretely, the teacher agents are trained to output $Q$-values satisfying the Bellman equation:

$$Q(s, a) = R^{s,a} + \gamma_{\text{Bellman}} \max_{a'} Q(s', a').$$ (2)

Thus the expected future reward is composed of the immediate reward, $\mathcal{R}^{s,a}$, of the action, $a$, and the maximum reward the agent can expect from the next state $s'$ onward when behaving optimally, i.e., picking the action that promises the most reward. The temporal discount $\gamma_{\text{Bellman}} \in [0, 1]$ signifies the uncertainty about rewards obtained for future actions ($\gamma_{\text{Bellman}} = 0$ would be maximum uncertainty, we use $\gamma_{\text{Bellman}} = 0.99$, see Supplementary Table S2).

To train the DQN, we minimize Mean Squared Error loss between the left- and right-hand sides of eq. (2), i.e., we minimize

$$\mathcal{L}_{\text{DQN}} = \frac{1}{|\mathcal{T}|} \sum_{\langle s, a, R^{s,a} \rangle \in \mathcal{T}} |Q(s, a) - (R^{s,a} + \gamma_{\text{Bellman}} \max_{a'} Q(s', a'))|^2,$$ (3)

where $\mathcal{T}$ is a set of transitions (state, action, and corresponding reward) $\langle s, a, R^{s,a} \rangle$ that the teacher DQN is trained on in the current optimization step. Thus, one optimization step is performed after each step the agent takes in the maze. The transition set $\mathcal{T}$ is composed of two distinct transitions: i) "long-term memory" transitions, which are all unique transitions the agent has seen since training began, and ii) additionally weighted "short-term memory" transitions, which are the last $L$ transitions the agent has seen. Therefore, the transitions that have recently been executed several times have a higher impact on the loss function $\mathcal{L}_{\text{DQN}}$ than the ones that were encountered a long time ago.

### Network specifications
The student and teacher networks are identical multilayer perceptrons apart from the input dimension. Each neuron in the two networks (except for the $K$ message neurons) has a ReLU activation function and a bias parameter. The number of parameters per layer for the student and teachers is listed in Table 3. The autoencoder neural network, which we use as a language proxy, consists of convolutional layers in addition to the fully connected layers. We use convolutions because the entries of the Q-matrix represent the states of the two-dimensional grid-world and, therefore, include spatial information that the network needs to learn. Thus, the input (Q-matrix of the teacher) is processed by two convolutional layers in the first half of the autoencoder, followed by one fully connected linear layer that outputs the message vector. After this dimensionality reduction, the decoding half of the autoencoder aims at reconstructing the original Q-matrix from the message vector. The architecture of the autoencoder is summarized in Table 4. For a brief study of the effect of different hyperparameters see Supplementary Figs. S11 and S12.

### Training and test tasks
The square grid-world setting consists of a grid of size $n \times n$ (see examples in Figs. 6, 7). Given that impenetrable walls surround each maze, this gives the agent an effective number of possible states (including the initial state where the agent starts, the goal state the agent has to reach, and the wall states the agent can not cross) equal to $\tilde{n} \times \tilde{n}$, where $\tilde{n} = n - 2$. In all cases, the agent starts in the bottom left corner. During the training of the SAE and student, we only include mazes with zero and one interior wall state, which gives us $(\tilde{n}^2 - 1) + (\tilde{n}^2 - 1)(\tilde{n}^2 - 2) = (\tilde{n}^2 - 1)^2$ possible maze-solving tasks. The agent moves through the grid-worlds with four discrete actions: single steps to the right, up, left, and down. Each episode starts with the agent at the initial state and ends when the goal is reached, or the maximum number of steps has been taken. To avoid potential infinite loops or movements into the walls, the agent receives a small negative reward for any action ($R_{\text{step}} = -0.1$) and a large negative reward for hitting any wall ($R_{\text{wall}} = -0.5$). If the agent reaches the goal, they receive a large positive reward ($R_{\text{goal}} = 2$).

**Table 3 | Network architecture in the teacher and student networks used in our toy model—in this context, $K'$ is the number of extra network inputs in addition to the state's x- and y-coordinates**

| layer | neuron number | weight parameters | bias parameters | total parameters |
|---|---|---|---|---|
| input layer | $2 + K'$ | – | – | – |
| linear layer | 10 | $20 + 10K'$ | 10 | $30 + 10K'$ |
| linear layer | 20 | 200 | 20 | 220 |
| linear layer | 20 | 400 | 20 | 420 |
| output layer | 4 | 80 | 4 | 84 |
| total ($K' = 0$) | 56 | 700 | 54 | 754 |
| total ($K' = 5$) | 61 | 750 | 54 | 804 |

This $K'$ corresponds to the length of the message, i.e., $K' = K = 5$ for the student, while the teacher does not receive a message, so $K' = 0$.

**Table 4 | Network architecture in the autoencoder network used in our toy model**

| layer | neuron number | weight parameters | bias parameters | total parameters |
|---|---|---|---|---|
| input layer | $4\bar{n}^2$ | – | $4\bar{n}^2$ | $4\bar{n}^2$ |
| conv. layer | 10 filters (size $2 \times 2 \times 4$) | 160 | 10 | 170 |
| conv. layer | 10 filters (size $2 \times 2 \times 10$) | 400 | 10 | 410 |
| linear layer | $K$ | $10(\bar{n}+2)^2 K$ | $K$ | $10(\bar{n}+2)^2 K + K$ |
| linear layer | $10(\bar{n}+2)^2$ | $10(\bar{n}+2)^2 K$ | $10(\bar{n}+2)^2$ | $10(\bar{n}+2)^2(K+1)$ |
| deconv. layer | 10 filters (size $2 \times 2 \times 10$) | 400 | 10 | 410 |
| deconv. layer | 4 filters (size $2 \times 2 \times 10$) | 160 | 4 | 164 |
| output layer | $4\bar{n}^2$ | – | $\bar{n}^2 4$ | $\bar{n}^2 4$ |
| total ($\bar{n}=4$ and $K=5$) | 493 and 34 filters | 4720 | 527 | 5247 |

In this context, $\bar{n}$ is the maze dimensionality (we use $4 \times 4$ mazes, therefore, $\bar{n}=4$) and $K$ is the length of the message (we use $K=5$).

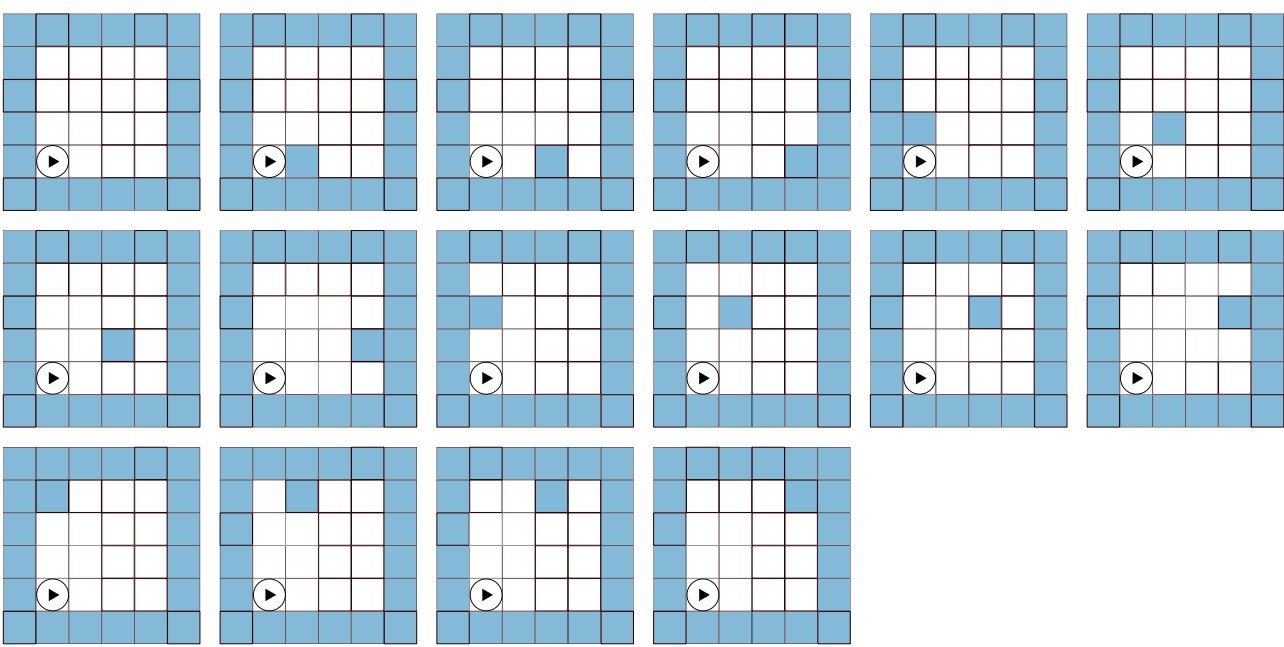

**Fig. 6 | All training mazes.** The student always starts in the bottom left corner. Light blue squares mark wall locations, which can not be accessed. The $4 \times 4$ mazes with 0 or 1 wall state comprise the training tasks.

We used all $4 \times 4$ grid-worlds with 0 or 1 wall state, amounting to 16 worlds in total, see Fig. 6 as tasks for training the language and the student. In world 0 (top left), there are 15 possible tasks, i.e., goal locations, namely all states that are neither a wall nor the starting location (all white squares without inset in the figure). Similarly, in the 15 worlds with a single wall, there are 14 possible tasks, amounting to 225 tasks used for training the language and the student agent. During the teacher training, the $Q$-values of the wall state positions of the teacher Q-matrix are set to 0, as the agent can never visit them (due to bounce back).

For unknown tasks, we chose all possible configurations of mazes with two wall states, six examples of which are shown in Fig. 7. We eliminated mazes that led to inaccessible states, leading to 101 possible configurations with two walls, each with 13 goal locations. Therefore, the test set was made up of 1313 test tasks in total.

**The full autoencoder loss**

The loss function for the SAE (which does not include student feedback) is defined as:

$$\mathcal{L}_{\text{SAE}} = \mathcal{L}_{\text{reconstruction}} + \mathcal{L}_{\text{sparsity}} = (1-\kappa)||\bar{Q} - Q||_2 + \kappa ||m||_1, \quad (4)$$

where $\kappa$ is a hyperparameter, which we can adjust to increase the importance of either the reconstruction or sparsity, $Q$ is the input Q-matrix, $Q'$ the reconstruction of the autoencoder, and $m$ is the lower-dimensional message.

We also included the student in the training of the language. Therefore, we augmented the autoencoder loss to include a term that enforced the usefulness of the messages to the student. This was done by first generating the student output for each possible state $s=(x_s, y_s)$, which consisted of four real numbers representing the four possible actions. Applying a softmax function to those values, we obtained action probabilities for the four actions in each state. Given all the action probabilities, we could calculate the state occupancy probabilities for the student after any number of steps $k$. We then defined the solve rate of a task as the state occupancy probability of the goal state after $k$ steps, as this state could not be left once it had been reached. We aimed for optimal solutions to be found; therefore, we always allowed the student only $k = k_{\text{opt}}$ steps to solve the task during training, where $k_{\text{opt}}$ was the length of the shortest path to the goal.

This amounts to the first term in eq. (5) of the student goal finding loss. The exponent was chosen to avoid the local minimum of the loss in which a small number of training tasks are not solved at all while the majority is solved perfectly. The second term in eq. (5) is a

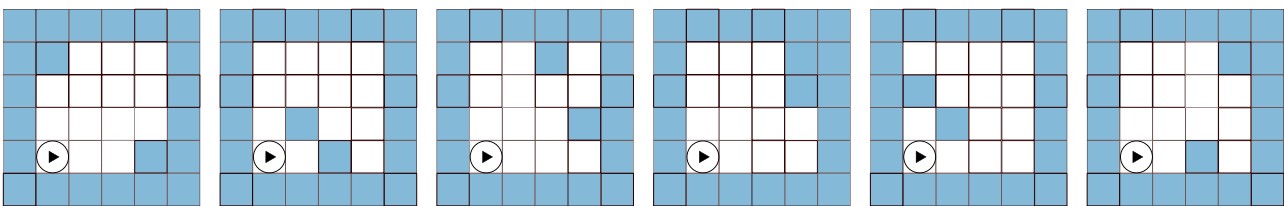

**Fig. 7 | Example test mazes.** The student always starts in the bottom left corner. Light blue squares mark wall locations, which can not be accessed. The $4 \times 4$ mazes with 2 wall states, except those where permissible states are cut off by walls, comprise the test tasks.

regularization of the student output while the hyperparameter $\gamma$ controls the relation between the two parts. The regularization of the student Q-matrix is also normalized by the number of its entries $4\tilde{n}^2$.

$$\mathcal{L}_{\text{goal finding}} = (1 - \gamma)(1 - \mathbb{P}[s_k = s_{\text{goal}}])^4 + \gamma \frac{||Q_{\text{student}}||_2}{\sqrt{4\tilde{n}^2}} \quad (5)$$

## Analysis of variance in the message spaces

We analyze the structure of the different message spaces by studying the relative variances explained by the two features describing each navigation task: the placement of the walls and the goal's location.

In this context, two types of variance can be computed: a *variance within groups* and a *variance between groups*, where a group is made up of either all tasks within a maze (i.e., same wall position) or all tasks with the same goal location. The former variance is lower when each group is clustered tightly, but the distance between groups is large. The latter variance is lower when the means of the groups cluster tightly, but there is a larger data spread within each group. To simplify the equations that follow, we introduce $M$, the total number of messages, $N$, the number of distinct groups, $M_i$, the number of elements in group $i$ and $m_{ij}$, which refers to the $j$-th message of group $i$. Then, the mean of each group is $\bar{m}_i = \frac{1}{M_i}\sum_{j=1}^{M_i} m_{ij}$ and the overall mean is $\bar{x} = \frac{1}{M}\sum_{i=1}^{N}\sum_{j=1}^{M_i} m_{ij}$. Thus the variance within and between groups of messages is defined by

$$\text{Var}_{\text{within}}(X) = \sum_{i=1}^{N}\sum_{j=1}^{M_i} (m_{ij} - \bar{m}_i)^2 \quad (6)$$

$$\text{Var}_{\text{between}}(X) = \sum_{i=1}^{N} M_i(\bar{m}_i - \bar{m})^2 \quad (7)$$

$$\beta = \frac{\text{Var}_{\text{between}}}{\text{Var}_{\text{within}} + \text{Var}_{\text{between}}} \quad (8)$$

Here, we introduce a value $\beta$, which allows for a comparison between the two different variances. When $\beta$ is close to one, the variance between groups dominates and vice versa.

Using the above variances, we can statistically test whether the means of all message groups (grouping either by wall position or goal location) are significantly different from each other by introducing the concept of the $F$-value, which is defined as the ratio of the mean square distance between groups $MS_B$ and the mean square distance within groups $MS_W$:

$$MS_B = \frac{\text{Var}_{\text{between}}}{N - 1}, \quad (9)$$

$$MS_W = \frac{\text{Var}_{\text{within}}}{M - N}, \quad (10)$$

$$F = \frac{MS_B}{MS_W}. \quad (11)$$

The group means differ significantly when the $F$-value is greater than a critical F-statistic (depending on a significance threshold $p$ and the degrees of freedom). As we removed the world with no wall states in the analysis of variances, the values of $F_{\text{crit}}$ (listed in Table 5) are the same in both grouping cases (by maze and by goal). The two degrees of freedom are $N - 1 = 14$ and $M - N = 195$.

## Statistical methods

One-sided $t$-tests were used when comparing the informed and misinformed students against the smart random walker, and a two-sided $t$-tests when comparing against the misinformed student. When multiple groups were considered, a Bonferroni factor was included in the t-tests.

## Language filtering used in the close-the-loop protocol

Initially, 25 languages were trained and evaluated when the student information was encoded and used for the navigation maze in Fig. 5. However, within that set of languages, we encountered a subset of languages (~30%) that led to lower solving rates for the informed student than the misinformed student or random walker on the trained tasks. The structure of these languages, which we defined as inefficient, led to a loss of task-critical information during the encoding. These languages were only removed from the set of languages we analyzed in Fig. 5. This language filtering was performed by retaining languages that, on average, led to a higher average task-solving rate for the informed student (receiving the message from encoded student information) compared to the average solving rate of the misinformed student and the random walker (all measured on the trained tasks). We included the misinformed student in the criterion to test whether our language is dysfunctional, i.e., the correct message leads to worse performance than a random message. Additionally, we included the random walker in this criterion as it is inherently intuitive that the informed student should perform better than the misinformed student, but that the language may nonetheless not provide a competitive advantage over taking random actions. Therefore, we check (i) if the

**Table 5 | Critical $F$-values for our data groupings by wall and goal for different significance thresholds**

| Significance threshold $p$ | Critical $F$-value $F_{\text{crit}}$ |
| --- | --- |
| 0.1 | 1.54 |
| 0.05 | 1.74 |
| 0.01 | 2.17 |
| 0.005 | 2.35 |
| 0.001 | 2.74 |

The statistical test performed was a one-sided F-test for variance analysis of groups.

**Table 6 | Exact *p*-values obtained using a two-sided and one-sided *t*-test (multiple tests were accounted for using a Bonferroni correction factor) for the informed vs misinformed and smart random walker, respectively in the results of Figs. 4, 5**

| Figure | | Maze scenario | p-val (Informed vs. Misinformed) | p-val (Informed vs. Smart Random) |
|---|---|---|---|---|
| Fig. 4 | e | i–vii | <0.0005 | <0.0005 |
| | f | ii | <0.0005 | <0.0005 |
| | | iii | <0.0005 | <0.0005 |
| | | iv | n.s. | n.s. |
| | | v | 0.025 | n.s. |
| | | vi | 0.014 | n.s. |
| | | vii | 0.001 | n.s. |
| | g | i–vii | <0.0005 | <0.0005 |
| | h | ii | <0.0005 | <0.0005 |
| | | iii | <0.0005 | <0.0005 |
| | | iv | n.s. | n.s. |
| | | v | 0.001 | n.s. |
| | | vi | 0.003 | n.s. |
| | | vii | <0.0005 | 0.020 |
| Fig. 5 | b | i | <0.0005 | n.s. |
| | | ii | <0.0005 | <0.0005 |
| | | iii | 0.001 | 0.008 |
| | | iv | <0.0005 | <0.0005 |
| | | v | 0.003 | <0.0005 |
| | | vi | <0.0005 | <0.0005 |
| | | vii | 0.016 | <0.0005 |
| | c | ii | 0.001 | n.s. |
| | | iii | 0.004 | 0.006 |
| | | iv | n.s. | n.s. |
| | | v | 0.023 | n.s. |
| | | vi | n.s. | n.s. |
| | | vii | n.s. | n.s. |
| | d | i | <0.0005 | n.s. |
| | | ii | <0.0005 | 0.001 |
| | | iii | 0.001 | n.s. |
| | | iv | <0.0005 | <0.0005 |
| | | v | 0.004 | 0.004 |
| | | vi | <0.0005 | <0.0005 |
| | | vii | 0.010 | 0.003 |
| | e | ii | <0.0005 | n.s. |
| | | iii | 0.002 | n.s. |
| | | iv | n.s. | n.s. |
| | | v | n.s. | n.s. |
| | | vi | n.s. | n.s. |
| | | vii | 0.025 | n.s. |

language is inherently functional and (ii) if it provides information that the student can use. The languages that fail our criterion are those that lead to task information loss or did not imbue the student with generalization abilities. Mathematically, we retain a language if

$$\mathbb{E}\left[Perf(S_I)\right] > \max(\mathbb{E}\left[Perf(S_M)\right], \mathbb{E}\left[Perf(\text{Random Walker})\right]) \quad (12)$$

where $\mathbb{E}$ is the expected value over all trained tasks, $S_I$ refers to the informed student who is provided the correct distorted message and $S_M$ is the misinformed student. We argue that this can be viewed through the biological lens where selective pressures favor more adaptive or efficient systems. Akin to the effect of natural evolution, where weak and inefficient members (in this case, languages) die out, languages that are detrimental to the student do not survive. For completeness' sake, we provided a supplemental figure where the full set of languages is displayed in Supplementary Fig. S13. As expected we see a decrease in the performance of the informed student for all scenarios but nonetheless, in all cases where previously the informed student performed better than the random walker/misinformed student, that ordering is maintained, i.e., even using the sub-optimal language embeddings allows the informed student to perform better at the tasks.

## Reporting summary
Further information on research design is available in the Nature Portfolio Reporting Summary linked to this article.

## Data availability
Source data are provided with this paper and can be found to generate the figures in this work can be found in the following public GitHub repository: github.com/meggl23/multi_agent_language, *zenodo.org/doi/10.5281/zenodo.7885526*[79].

## Code availability
Computer code to train the agents, generate languages, and plot the figures can be found in the following public GitHub repository: github.com/meggl23/multi_agent_language, *zenodo.org/doi/10.5281/zenodo.7885526*[79].

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

## Acknowledgements
We acknowledge the support of the Institute of Experimental Epileptology and Cognition Research at the University of Bonn Medical Center and the Joachim Herz Foundation (M.F.E. and C.W-C.). This research was funded by the Deutsche Forschungsgemeinschaft (DFG, German Research Foundation)—Project-ID 227953431—SFB 1089 (T.T.). We thank Alison Barker and Martin Fuhrmann for fruitful discussions, and all members of the Tchumatchenko group, particularly Pietro Verzelli, for feedback on the manuscript. This work was supported by the Open Access Publication Fund of the University of Bonn.

## Author contributions
T.J.W., experimental design, code writing, data analysis, visualization, and writing; T.T., supervision, project administration, experimental design, funding, and writing; C.W-C., conceptualization, supervision, experimental design, data analysis, and writing; M.F.E., conceptualization, supervision, experimental design, data analysis, and writing.

## Funding

## Competing interests
The authors declare no competing interests.
