## [Peer Review File · Nature Communications]

A framework for the emergence and analysis of language in social learning agentsREVIEWER COMMENTS

Reviewer #1 (Remarks to the Author):

The main result is an experimental framework for studying the emergence of communication protocols with navigation tasks in a grid world. The communication interface is implemented as a sparse autoencoder, creating a bottleneck between two neural network modules. Experiments compare passing a message corresponding to the correct task against passing a message corresponding to a wrong task. The correct message case leads to substantially higher success rates, indicating that the receiving module does make use of the message.

Significance

There is rich literature (non-exhaustive reference list at the end) on emergent communication (mostly referential games, but also grid-world navigation games as in the manuscript). Most of this literature, models the communication interface as a *discrete* message (trained via REINFORCE and/or Gumbel Softmax). The manuscript proposes to model the communication interface as a sparse autoencoder, but the transferred message is still a continuous message and the gradient flows through the code. The manuscript would benefit from a more detailed discussion of advantages and disadvantages of their approach compared to other approaches, e.g., with a discrete communication interface, reflecting on the usefulness of the approach for simulating and studying language emergence. I am concerned that the impact of this work could be limited given the existing literature, but I am hopeful that this can be sorted out in a revision.

Does the work support the conclusions and claims?

While I strongly appreciate the conservativeness in discussing the results, I feel the paper could be strengthened by stating the claims as well as theoretical and practical implications more clearly. Assuming the main claim is that the proposed experimental framework is a useful tool to study the evolution of language, it would be necessary to better contextualize the work within the existing literature on emergent communication and computational modeling of language evolution and comparing their framework to others. For example, one claim is that the simulated language evolves according to a utility or gain function; which is not certainly not unique to this work. The discussion then draws some interesting connections to phenomena observed in animals and humans. However, these connections remain rather shallow: a) the similarity to non-primate mammals is based on having model-free agents b) the connection to phenomena in humans is solely based on the relationship between quality of representations and agents' performance. Instead, a suggestion for strengthening the paper would be to show that key features of language, such as a consistent form-to-meaning mapping or compositionality, do indeed emerge in the communication protocols. For instance, compositionality can be quantified by topographic similarity (Brighton & Kirby, 2006; Lazaridou et al., 2018). In addition, maybe an information-theoretic analysis of the message/code would lead to further insights (e.g, see Kharitonov et al., 2020). It also might be worth checking Lowe et al. (2019) to further enrich the analysis.

Data analysis, interpretation, and conclusions

I appreciate the transparency that 30% of the emergent languages were removed and not included in the presented results (Fig. 4). In pursuit of a reasoning, the manuscript resorts to natural evolution as an analogy, which is little convincing. While I understand that this might be an artefact of unstable reinforcement learning in general, I wonder to what extent the necessity of removing 30% of the results impacts the usefulness of the framework and the conclusions drawn from the experiments. The paper could be improved by adding a version of Figure 4 with **all** results as supplementary material. Additionally, the criteria for a resulting language to be included needs to be stated more clearly.

The manuscript repeatedly states that some higher dimensional meaning space is encoded into a lower dimensional message space, where the lower dimensional message space is supposed to correspond to language. This stands in contrast with the ideas of e.g., word2vec (Mikolov et al., 2013), which encodes the **high-dimensional** language data (think: alphabet size to the power of message length) into a low-dimensional, continuous representation. As the stated goal of the developed framework is to simulate and analyse the emergence of language, the notion/motivation assumed in the manuscript could be described more clearly. In other words, why should the communication channel be low dimensional?

Soundness of the methodology

The methodology seems to be for the most part sound. One issue is that the manuscripts state that they use an L2 penalty to promote sparsity. In contrast, it is common to use an L1 penalty to promote sparsity. In contrast, L2 penalty resembles a pressure towards a Gaussian prior, but not necessarily towards sparsity. I suggest to provide more background on why an L2 penalty should promote sparsity in a revised version of this manuscript.

Moreover, I'm wondering whether it is reasonable to consider teacher and student as separate agents. The two(?) agents are connected by a continuous channel **and** the gradient is propagated back through this channel. Unless I misunderstood some part of the approach, this would be equivalent to training a single neural network model (with a regularization term on a bottleneck module). If this was the case, it would render the main result little surprising: swapping out some intermediate representation with an intermediate representation corresponding to a different example (here: task), would certainly degrade the performance.

Reproducibility

The approach and hyperparameters are described in sufficient detail, yet the criteria for hyperparameter selection could be stated more clearly. A sensitivity analysis for the student-feedback weighting factor is provided. However, it could be complemented by analyzing varying other critical hyperparameters, such as the code dimension of the autoencoder. Lastly, for full reproducibility and reuse of the experimental framework, it would be necessary to share the source code.

References

Coming back to connecting with the literature: I explicitly do **not** request citing all of this literature but I encourage to look into it and discuss the relationship of the proposed approach to the general themes of a) (discrete) emergent communication and b) (cooperative) multi-agent reinforcement learning, and c) computational modeling in language evolution research. I hope this list is helpful.

- Chaabouni, R., Kharitonov, E., Bouchacourt, D., Dupoux, E., & Baroni, M. (2020). Compositionality and Generalization In Emergent Languages. ACL 2020.
- Chaabouni, R., Kharitonov, E., Dupoux, E., & Baroni, M. (2021). Communicating artificial neural networks develop efficient color-naming systems. *Proceedings of the National Academy of Sciences*, 118(12).
- Chaabouni, R., Strub, F., Alché, F., Tarassov, E., Tallec, C., Davoodi, E., Mathewson, K. W., Tieleman, O., Lazaridou, A., & Piot, B. (2022). Emergent communication at scale. *Proceedings of ICLR*.
- Foerster, J., Assael, I. A., de Freitas, N., & Whiteson, S. (2016). Learning to Communicate with Deep Multi-Agent Reinforcement Learning. *Advances in Neural Information Processing Systems*, 29.
- Galke, L., Ram, Y., & Raviv, L. (2022). Emergent Communication for Understanding Human Language Evolution: What's Missing? *Emergent Communication Workshop at ICLR*.
- Gong, T., Shuai, L., & Zhang, M. (2014). Modelling language evolution: Examples and predictions. *Physics of Life Reviews*, 11(2), 280–302.
<https://doi.org/10.1016/j.plrev.2013.11.009>
- Guo, S., Ren, Y., Mathewson, K. W., Kirby, S., Albrecht, S. V., & Smith, K. (2021, October 6). Expressivity of Emergent Languages is a Trade-off between Contextual Complexity and Unpredictability. *International Conference on Learning Representations*.
https://openreview.net/forum?id=WxuE_JWxjKw
- Havrylov, S., & Titov, I. (2017). Emergence of Language with Multi-agent Games: Learning to Communicate with Sequences of Symbols. *Advances in Neural Information Processing Systems*, 30.
- Kharitonov, E., Chaabouni, R., Bouchacourt, D., & Baroni, M. (2020). Entropy Minimization In Emergent Languages. *Proceedings of ICML*.
- Kirby, S., & Tamariz, M. (2022). Cumulative cultural evolution, population structure and the origin of combinatoriality in human language. *Philosophical Transactions of the Royal Society B: Biological Sciences*, 377(1843), 20200319.
<https://doi.org/10.1098/rstb.2020.0319>
- Lazaridou, A., Hermann, K. M., Tuyls, K., & Clark, S. (2018). Emergence of Linguistic Communication from Referential Games with Symbolic and Pixel Input. *Proceedings of ICLR*.
- Li, F., & Bowling, M. (2019). Ease-of-Teaching and Language Structure from Emergent Communication. *Advances in Neural Information Processing Systems*, 32.
- Lowe, R., Foerster, J., Boureau, Y.-L., Pineau, J., & Dauphin, Y. (2019). On the Pitfalls of Measuring Emergent Communication. *AAMAS 2019*. <http://arxiv.org/abs/1903.05168>
- Michel, P., Rita, M., Mathewson, K. W., Tieleman, O., & Lazaridou, A. (2023, February 1). Revisiting Populations in multi-agent Communication. *Proceedings of ICLR*.
- Ndousse, K. K., Eck, D., Levine, S., & Jaques, N. (2021). Emergent Social Learning via Multi-

agent Reinforcement Learning. Proceedings of ICML.

- Oroojlooy, A., & Hajinezhad, D. (2022). A review of cooperative multi-agent deep reinforcement learning. *Applied Intelligence*, 1-46.
- Rita, M., Strub, F., Grill, J.-B., Pietquin, O., & Dupoux, E. (2021, September 29). On the role of population heterogeneity in emergent communication. *Proceedings of ICLR*.
- Ruis, L., Andreas, J., Baroni, M., Bouchacourt, D., & Lake, B. M. (2020). A benchmark for systematic generalization in grounded language understanding. *Advances in Neural Information Processing Systems*, 33.
- Tieleman, O., Lazaridou, A., Mourad, S., Blundell, C., & Precup, D. (2019). Shaping representations through communication: Community size effect in artificial learning systems. *NeurIPS 2019 workshop on visually grounded interaction and language*.
- Tucker, M., Li, H., Agrawal, S., Hughes, D., Sycara, K., Lewis, M., & Shah, J. A. (2021). Emergent Discrete Communication in Semantic Spaces. *Advances in Neural Information Processing Systems*, 34.

Reviewer #3 (Remarks to the Author):

This paper provides a nice and interpretable framework/model that allows language to emerge in multi-agent systems allowing for the analysis of RL agents that are equipped with the ability to represent and communicate features of an artificial language that directly involves the task they are trained to perform. They show that for tasks represented in teacher/student setting, the student learns to decode messages from the teacher to perform tasks. When measuring % tasks solved, they see that informed students (that get message information) perform significantly better than misinformed students or random walkers, however student-student performance leads to lower performance sometimes. They also analyse the embeddings produced by each model to visualise the differences in representations learned.

I think a lot of the analyses could be framed better to allow a better understanding of the different components of the system. Specific points below:

Some analyses do not seem to be in line with the rest of the results/hypotheses posited here. For example, the PCA plots can allow us to understand if there is a difference in the language embeddings after training and whether embedding cluster in intuitive ways, however the results don't seem very conclusive. It would be good to have a discussion section that elaborates on these, and also a more thorough investigation of different types of dimensionality reduction approaches that might lead to clear results.

There is also the question of why the autoencoder architecture is important to this setup—most other papers try to show results that are model/architecture-agnostic (e.g., this multi-agent communication setting should work with an RNN/LSTM/Transformer or any model with RL feedback) so it would be good to see that these hold up for other architectures and we can make the same claims about language/communication.

Line by line comments:

1. Page 1: "Language in a broad sense (citation 9)" → it is unclear what this means given the citation is about the use of language in grounded situations and with a communicative intent, rather than what this sentence implies it is being cited for. It might be worth rephrasing the sentence here to correctly reference the cite mentioned.
2. "Surge of papers that attempt to combine linguistic properties with deep neural network..." → a better rephrasing is "papers that combine multi-agent systems with communication policies." The current sentence is slightly incorrect since the current phrasing (linguistic properties + deep learning) is more applicable to the entire field of NLP rather than just the emergent communication multi-agent work that the authors should be citing here
3. The last paragraph before "model architecture" can be written better to concretely summarise the goals. Maybe a bullet point list of the main contributions of this paper would be helpful to make this paper clearer.
4. "Natural language is a lower dimensional representation of higher level concepts" → cite to Manning et. al., does not seem very well founded. The textbook cited gives an overview of statistical NLP/parsing/other algorithms, but is not a good source of the sentence written here. It is not a widely held hypothesis that natural language is "lower dimensional" than any concepts, so if the authors want to claim this it would be worth arguing this with the correct citations.
5. "Ought to be generalisable" → it would be nice to frame this by drawing an analogy to humans or our expectation of why we would want task generalisation.

Dear referees,

Thank you for your invitation to revise our manuscript, the helpful feedback, and the insightful comments on our manuscript that helped improve the accessibility of the work. Below, we provide a point-by-point reply listing the original comments (in bold) and our reply in black (not bold) with direct quotations from the revised manuscript in italics. Changes in the updated manuscript are additionally highlighted in red.

REVIEWER COMMENTS

Reviewer #1 (Remarks to the Author):

The main result is an experimental framework for studying the emergence of communication protocols with navigation tasks in a grid world. The communication interface is implemented as a sparse autoencoder, creating a bottleneck between two neural network modules. Experiments compare passing a message corresponding to the correct task against passing a message corresponding to a wrong task. The correct message case leads to substantially higher success rates, indicating that the receiving module does make use of the message.

Significance

There is rich literature (non-exhaustive reference list at the end) on emergent communication (mostly referential games, but also grid-world navigation games as in the manuscript). Most of this literature, models the communication interface as a *discrete* message (trained via REINFORCE and/or Gumbel Softmax).

The manuscript proposes to model the communication interface as a sparse autoencoder, but the transferred message is still a continuous message and the gradient flows through the code.

We would like to thank the reviewer for the appreciation of our work and for the opportunity to clarify the novelty aspects in relation to previous work, especially regarding the nature of the representational space.

Indeed, we model inter-agent communication by considering the internal representation of continuous variables within the neural network of each agent, compressing it into a message, and propagating it to another agent (student). We believe this configuration is best positioned to address the aims of our article: analyze and relate the internal, external, and social representations of agents in tasks and obtain a direct link between task performance and the representational spaces, including the compressed message space that serves bidirectional communication. Underlying all these goals is a minimalistic setup. By relying on a continuous close-to-linear system (in our case, a sparse autoencoder), we can meet the requirements of a condensed message space for communication while doing the same analysis in the message state-space and the agent state-space. Therefore, we can establish direct connections between the environmental variables that are propagated to the language space and are being captured by the agent by construction. In contrast, relying on a discrete or symbolic space would remove the dimensionality and statistical study from the other two spaces.

We agree with the reviewer that existing communication research often focuses on discrete protocols with fixed vocabularies. Our approach and results, in contrast, are motivated by the notion that words, vocalizations, or, in general, certain semantic hierarchies, though consisting of discrete items, represent continuous concepts (e.g., object positions in a maze) and that these words ultimately carry signatures of the neural representation of these object hierarchies in the brain. For example, MacKenzie et al. (<https://www.sciencedirect.com/science/article/pii/S089662731400405X>) showed that hierarchically organized neural schemas in the hippocampal circuits of the brain correspond to object/word hierarchies and recent work by Safaie et al. showed that neural representations across individuals of the same species can be mapped to each other (<https://www.nature.com/articles/s41586-023-06714-0>). This neuroscientific perspective underlies our understanding of the message space. Let us note that prior research centering on the concept of fixed vocabularies was beneficial for studying modularity and semantics. However, these differ from natural languages, which evolve through learning, adapt to contexts, and seem to have a neural correlate in the brain. In natural language, we can describe varied shades of color and create new meanings, like "green light" for "go" or "red herring" for a distraction, and form semantic hierarchies that seem to correlate with neural activity (e.g., MacKenzie et al.). Language's dynamic and flexible nature is evidenced by the annual addition and removal of words in dictionaries, highlighting the yet-to-be-achieved flexibility and continuity in state-of-the-art synthetic language protocols.

We have adapted the introduction to enhance the accessibility and our motivation for how neural representation and the message space are related:

"To understand the interplay between environmental experiences and their abstractions, we employ a reinforcement learning (RL) framework²⁹. This approach is rooted in the idea that task abstractions are empirical and vary among agents. We are particularly interested in three aspects: how agents internally abstract real-world variables, how these abstractions translate into a common, shareable language, and the interaction of these elements. Hence, we opted for a non-discrete language model to directly compare

the continuous nature of both brain processes and real-world phenomena. By feeding into the language model the learned information provided by RL agents performing a navigational task, we investigated the development of natural language as it arises from social and decision-making processes³⁰. This leads to individualized abstractions emerging organically rather than being pre-defined, in contrast to supervised learning methods^{27, 31, 32}. By analyzing the structure of the language embedding, we can gain insights into information content in the message space and its relation to neural representations underpinning task performance and generalization. Our methodology, involving continuous communication between a student and a teacher agent, allows us to study the emergence of message space structure and recapitulate natural language features, such as role reversal and language evolution³³. Additionally, it stands apart from previous non-supervised, symbolic methods^{34–38}, taking cues from continuous language generation models^{39, 40} and animal communication systems, like the bee’s waggle dance, which translate a continuous environment into a concise message space^{41, 42}, also seen in human languages^{39, 40}.

The manuscript would benefit from a more detailed discussion of advantages and disadvantages of their approach compared to other approaches, e.g., with a discrete communication interface, reflecting on the usefulness of the approach for simulating and studying language emergence.

I am concerned that the impact of this work could be limited given the existing literature, but I am hopeful that this can be sorted out in a revision.

We thank the reviewer for this helpful and encouraging comment. We are grateful for the list of additional literature to compare our work to and have added his/her list to the updated manuscript with an in-depth review of the previously existing approaches.

The main contributions of our paper are the insights into representations of the external world in the messages and the agent ANNs: We studied the co-development of the internal and language spaces simultaneously in relation to the cues in the external world. The representational space covered by our RL agents is continuous, self-generated, and can be used for symmetric communication. Below, we detail three points that we believe set our work apart:

- Insights into how the representational spaces are shared and communicated brain-to-brain or ANN-to-ANN. Several articles have highlighted that this challenge is a fundamental and open problem in neuroscience and artificial neural networks and needs attention (Hasson et al. 2012 and Tieleman et al. 2019). This facet is also related to studying the concept of theory of mind (ToM) (Premack & Woodruff, 1978) in a cooperative multi-agent paradigm as a communication protocol (Yuan et al 2020).
- Understanding how communication is enhanced by linking it to task performance. Without explicit incentives, the reconstruction loss of the auto-encoder is decreased when student performance is fed back to it. This leads to a more efficient message that the student can then subsequently take advantage of. This result is slightly unintuitive as you would argue that better language embedding would lead to better performance, while in our case, we observe that better performance leads to a better language embedding.
- We provide a possible avenue to studying symmetric communication, recently highlighted by Galke et al. 2022 as an important challenge in the field. Historically, the sender and the receiver had distinct, non-interchangeable roles. However, a common message and representational space in brain-to-brain or ANN-to-ANN communication relies on the fact that any agent should be able to send and understand the meaning of a message and modify it if necessary.

Below, we discuss each of these three key novel points our work provides and have sharpened the introduction and our paper overall to make this more readily accessible to our readers. As suggested by another reviewer, we have added these contributions to the introduction to highlight the impact of our work to the introduction:

“To summarize, we present a tractable framework for understanding emergent communication, disentangling the relation between the internal neural representations and the message space that contributes following three novel results to the neuroscience and neuroAI communities:

- *reveal features in the lower-dimensional embedding space necessary for higher student success and task generalization^{6,43}.*
- *demonstrate that the communication is enhanced and the information content of the message increased when the communication channel is provided with feedback in the form of the student’s task performance.*
- *provide an avenue to studying symmetric communication, i.e., where the sender and receiver can be interchanged, highlighted recently as an important challenge in the field³³.*

“

Additionally, we agree that a comparison of our work to existing literature would emphasize the usefulness of our framework.

We were inspired by the work of Foerster et al. 2016 (whom we cite in the manuscript) that introduced the concept of “differentiable inter-agent learning” (DIAL) and reinforced inter-agent learning (RIAL) (see Tampuu et al. 2017 for another application). RIAL is defined by separate agents where each agent learns their network parameters with a discrete message channel, treating the other

agents as part of the environment, while DIAL uses real-valued messages (predefined and finite in number) being passed between agents (treated as one network) during centralized learning, leading to the communication acting as a bottleneck connection between agents. In our work, we use the real-value communication channel of DIAL in combination with the separate agents of RIAL. Concretely, this means that the teacher and student share the communication protocol only in a feedforward manner; gradient information does not propagate backward.

Our work also includes ideas linked to the work of Tieleman et al. 2019, which we have now amended in the updated version. Tieleman et al. 2019 used an encoder-decoder structure to study the effect of community size on the representations that agents form to communicate. They considered auto-encoders that derive from multiple agents but did not study the resulting lower-dimensional representations in a task setting. We similarly used an auto-encoder to generate lower-dimensional representations but also studied these in the context of task completion of the agent receiving that representation.

Additional work whose spirit is related to our work is Tucker et al. 2021, which uses a continuous message space. The authors wrote that “discretizing messages by constraining them to conform to one-hot vectors fundamentally precludes agents from learning some desirable properties of language” and that using one-hot vectors, which are naturally orthogonal to each other and equally far away from all other vectors, prevents the establishment of relationships between tokens. Given that objects in space, and by extension, natural language, can have arbitrary orthogonal relations to each other, using a continuous real-value communication channel, as is done in our work, can overcome this limitation. Additionally, research from natural language processing has long established the importance of learning representations of discrete words within a continuous space (see, for example, Mikolov et al., 2013).

Therefore, we believe that our approach differs sufficiently from the existing literature because it provides novel insights and advantages. It goes beyond the work of Tielman et al. 2019 and Tucker et al. 2021 by studying the generalization of our language and how the structure of the message space affects this. Additionally, we can generate certain features of natural language, such as the role reversal of the student or the evolution/perturbation of a language, that may be challenging with a discrete message space. Also, the study of the representational content and flow from teacher to message and ultimately to the student is novel with respect to existing literature and has been highlighted as an open challenge by several studies, including Hasson et al. 2012.

We have added this comparison to the discussion of the manuscript as follows:

“Drawing inspiration from Tucker et al.³⁷, our research builds upon their findings that agents can effectively communicate in noisy environments by clustering tokens within a learned, continuous space. Additionally, we reference the work of Foerster et al.¹⁵, who developed a model for independent parameter learning by agents, along with a system facilitating real-valued messages during centralized learning. Unlike the approach of Foerster et al., which shares gradient information between agents for end-to-end training, our method uses a continuous channel solely for task representations and trains agent parameters separately, without shared gradient data. This approach yielded a latent structure that prioritized variability along the goal space instead of the maze configuration, contrasting with the prominence of the state space in solely teacher-based models.”

Let us mention that we did not compose messages by concatenating words to generalize our message, i.e., chaining together words to form new words, a key feature of natural language. Following the work of Kharitonov and Baroni, 2020, who noted that emergent language generalizations were not necessarily tied to compositionality, we decided instead to focus on the language being generalizable through interpolation. Nonetheless, given the critical nature of composability, future work consisting of a sequential set of tasks could extend this feature.

Secondly, by encoding task-relevant information, the lower-dimensional embedding space will, by definition, include features of that task information, which may lead to a bias in the generation of our language. This decision was taken to facilitate the study of the lower-dimensional representations but simultaneously simplifies the resulting language. Future work will extend the framework to encode different information (e.g., the activations of a hidden layer of the teacher) to understand what information is best to encode.

Finally, we acknowledge that discrete communication protocols are more common than the continuous approach we utilized in our work. While the continuous structure of our representations led to genuine insights, it was not straightforward to infer language-specific concepts directly, as the latent variables of the embedding were complex combinations of the task information. In contrast, a discrete vocabulary provides a more intuitive space and consists of dimensions that can directly be attributed to key information. Future work will include combining a discrete approach and our existing framework by projecting the messages from the auto-encoder onto a discrete set of messages. In this way, we can have the natural evolution of our language as the auto-encoder is trained, but the simplicity of a discrete communication channel.

We have summarized these points in brief paragraphs in the discussion as follows:

“In this study, we did not utilize sequential composition to generalize our message^{32, 68}. Instead, we aimed to generalize through interpolation of the continuous messages. Nonetheless, the framework can readily be extended to include composable messages using sequential sets of tasks, which will be the focus of future studies.”

[...]

“By encoding task-relevant information (e.g., Q-matrices), the lower-dimensional embedding space was biased to include features of the task information indirectly. This was observed in the emergent hierarchical latent structure obeying task variables and is similar to social species that show cultural or experience-dependent complexity in their linguistic traits, like non-primate mammals such as bottlenose dolphins⁷¹ or naked-mole rats⁷². In this sense, we presume that the neural representations and circuitry of the agents evolve and rewire to enable social learning⁵. By doing so, we look at the interplay between the community scale and the cognitive one instead of fixing communication or neuronal representations. Thus, research around generalist agents performing dual roles as teachers and students is crucial. This involves creating agents with distinct sender and receiver units and an experience-based policy. Additionally, examining the impact of the social graph on language construction and expanding to further tasks is vital. In future work and inspired by Tieleman et al.⁴³, who employed an encoder-decoder model to examine how community size influences message representations by additionally investigating receiver task performance, we aim to enhance the framework to encode different information (e.g., the activations of a hidden layer of the teacher). This would allow us to reverse the student-teacher roles as the information to be encoded is available in both agents and to understand whether this structure emerges through other embeddings and how this information propagates through the agents. Finally, discrete communication protocols are more commonly used than the continuous approach in our work^{34–38}. Future work will combine discrete messages and our existing framework by projecting the messages that arise from the auto-encoder onto a discrete message space to attempt further to emulate natural language.”

Does the work support the conclusions and claims?

While I strongly appreciate the conservativeness in discussing the results, I feel the paper could be strengthened by stating the claims as well as theoretical and practical implications more clearly.

Assuming the main claim is that the proposed experimental framework is a useful tool to study the evolution of language, it would be necessary to better contextualize the work within the existing literature on emergent communication and computational modeling of language evolution and comparing their framework to others.

For example, one claim is that the simulated language evolves according to a utility or gain function; which is not certainly not unique to this work. The discussion then draws some interesting connections to phenomena observed in animals and humans.

However, these connections remain rather shallow: a) the similarity to non-primate mammals is based on having model-free agents b) the connection to phenomena in humans is solely based on the relationship between quality of representations and agents' performance.

We thank the reviewer for encouraging us to improve the discussion of the results and implications of our study.

Firstly, we would like to highlight that our results have practical and theoretical consequences on how representational meaning is developed both at the level of internal and shared representations. Concretely, our model predicts that both follow a similar hierarchization as environmental or behavioral constraints, which, in turn, are guided by the environmental needs. How state-space is abstracted to follow goal variability more than inter-world structure changes – i.e., walls – points to this phenomenon. Following the advice of the reviewer, we have summarized our contributions at the end of the introduction:

“To summarize, we present a tractable framework for understanding emergent communication, disentangling the relation between the internal neural representations and the message space that contributes following three novel results to the neuroscience and neuroAI communities:

- *reveal features in the lower-dimensional embedding space necessary for higher student success and task generalization^{6,43}.*
- *demonstrate that the communication is enhanced and the information content of the message increased when the communication channel is provided with feedback in the form of the student's task performance.*
- *provide an avenue to studying symmetric communication, i.e., where the sender and receiver can be interchanged, highlighted recently as an important challenge in the field³³.*

“

We agree with the reviewer that the idea of model-free systems is arguable. However, this is the most assumption-free system when considering training accounting for both an evolutionary and within-system language emergence. In a model-based approach, we would have had to assume the state transitions of our spatial system, which would go against our naive neuronal approach. As the main focus of our work is to avoid any *a priori* considerations for the internal representations, we believe this was the most applicable approach to implement our system.

In the introduction, we added a more in-depth comparison with existing literature on emergent communications and computational modeling of language evolution. Additionally, we revised the discussion section to include further analysis of the implications of our

study. In particular, we discuss both how utility drives language evolution and how form-to-meaning mapping and compositionality emerge in different communication systems by animals.

Finally, we enriched the theoretical and experimental implications in the discussion section:

“The implications of our study suggest possible analogies with natural languages. First, our system evolves according to a utility or gain function, not solely to error minimization or comprehensibility. Lossless information transmission is insufficient for competent behavior, and the message space needs to adapt to be advantageous for other agents. This is similar to the natural language, where morphemes evolve according to motives, goals, and efficiency of a group^{30, 62}. For instance, in birds, it has been observed that utility drives the emergence of new linguistic relations or compositions^{63, 64}. Second, introducing dimensionality and sparsity constraints is motivated by anatomical and cognitive limitations, such as vocal tract size or memory capacity^{65, 66}. Hence, by allocating a predefined number of dimensions to our communication system, we replicate such properties and observe that these are organized into hierarchical task-relevant modes. However, ongoing work still aims to answer how channel size relates to the representational space, as machine learning and brain activity tend to converge to a high-dimensional space in the representations that are not shared by the actual symbolic space^{50, 67}. Studies have shown that brain activity is compressed relative to the message space even if our languages are not precisely low-dimensional^{10, 48}.”

Let us mention that the introduction of new model animals that exhibit strong social group behaviors, such as naked mole rats (as shown by Barker et al., 2021), opens new avenues for studying the relation between neural abstractions in the brain and language. In this regard, our approach enables synergies between scientific fields addressing agent-based decision-making tasks, communication protocols, and linguistic structure.

Instead, a suggestion for strengthening the paper would be to show that key features of language, such as a consistent form-to-meaning mapping or compositionality, do indeed emerge in the communication protocols.

For instance, compositionality can be quantified by topographic similarity (Brighton & Kirby, 2006; Lazaridou et al., 2018). In addition, maybe an information-theoretic analysis of the message/code would lead to further insights (e.g, see Kharitonov et al., 2020). It also might be worth checking Lowe et al. (2019) to further enrich the analysis.

We thank the reviewer for the comments and recommendations on improving our analysis, which we have included in our updated manuscript. As he/she suggested, we first performed a compositionality analysis using topographic similarity. To calculate the topographic similarity, we determined the relationship between the messages and (i) the task labels, i.e., the spatial difference in the mazes, and (ii) the information that the teacher provides to the auto-encoder (Q-matrices). Using a similar approach to that of Lazaridou et al. (2018), we then calculated the message distance (Euclidean norm) against meaning distances for the two different metrics. The distance in the labels was calculated as a weighted sum of the differences between the goal and the wall locations, while the distance of the teacher Q-matrices, which represents a combination of space-based and action-based meaning, was calculated using a Frobenius norm.

We find that the message distances and task label/teacher output meanings show topographic similarity as indicated by the positive slope parameter of the linear regressions (for both metrics). Taking this slope as a quantitative measure, we also find that languages with feedback show higher degrees of topographic similarity (and thus compositionality) than languages without feedback, which reinforces our hypothesis that hierarchization of the language depends on task demands or needs of the agent. Hence, language seems to be better at the agent-level performance-wise when it adapts itself according to some utility-driven hierarchy.

We also performed an information theoretic analysis using Shannon entropy to measure the information-carrying capabilities of the messages. As Shannon entropy is restricted to random variables taking discrete values and our messages arise from a continuous embedding space, we projected our messages onto a set of discrete bins and then calculated the entropy of that discrete distribution. While this discretization means that some information is lost, a direct comparison of the entropy of tasks (the maximum entropy value), teacher outputs, messages, and student output becomes possible. To verify that our findings are independent of the bin size we chose, we calculated the entropy across a variety of bin widths. Additionally, for visualization purposes, we only show the first two PCs and normalized the distributions with an equal factor along both PC-axes so that samples were restricted to $[-1, 1]^2$ in the depiction of the example binning.

We found that the entropy decreases when moving through the communication framework, i.e., the entropy of the teacher outputs is highest, followed by the entropy of the messages and finally the entropy of the student outputs. This result follows intuitively, given that at each stage, information is lost as it passes through an agent/network. We note that when student performance is provided to the auto-encoder during training, the resulting message and the student entropy are higher than when there was no feedback. This means the language becomes more effective in conveying information through this mechanism.

Additionally, we aimed to confirm one of the findings of Kharitonov et al., 2020, which states that there is pressure for a language to be as simple as possible and that this pressure is amplified as we increase communication channel sparsity. We simulate this scenario by removing the reconstruction loss from the auto-encoder training. In this scenario, the auto-encoder loss only consists of the sparsity promotion and the student performance feedback, which amplifies the pressure of the auto-encoder to generate a sparse

message space. We find that the entropy of the messages and student output is significantly lower than in the previous scenario, and the difference between the two (message and student action entropy) is not significant.

We have added a discussion to this effect to the results section of the updated manuscript, as well as the below figure, which depicts the described results:

Figure 3. Topographic similarity and entropy analysis of the emergent languages. *a)* Visualization of goal and wall distance vectors between two tasks (task 1: solid, task 2: checkered). In combination, these are used to compute a spatial task distance $\Delta t = ||R_{wall}\Delta w||_2 + ||R_{goal}\Delta g||_2$ for *b)*. *b), c)* Comparison of pairwise task meaning distances and pairwise distances of the corresponding messages. Message and task distances are measured with the Euclidean norm, matrix distances with the equivalent for matrices, the Frobenius norm. Standard deviations are computed over five languages (different random seeds). m refers to the gradient of the linear fit. *d)* Entropy analysis through discretization: First two PCs of normalized samples of each data type binned. The entropy is then computed for different choices of bin size. Here, we depict an example discretization for 5 bins in each PC direction (bin side lengths are identical in both directions). *e)* Calculating the entropy for each PC discretization demonstrates a clear ranking of the teacher, message and student information carrying capabilities. The maximum possible entropy is that of a uniform distribution over all maze tasks.

Finally, we are grateful to the reviewer for making us aware of Lowe et al. 2019 which we originally failed to cite in our manuscript, but now do.

Data analysis, interpretation, and conclusions

I appreciate the transparency that 30% of the emergent languages were removed and not included in the presented results (Fig. 4). In pursuit of a reasoning, the manuscript resorts to natural evolution as an analogy, which is little convincing.

While I understand that this might be an artifact of unstable reinforcement learning in general, I wonder to what extent the necessity of removing 30% of the results impacts the usefulness of the framework and the conclusions drawn from the experiments. The paper could be improved by adding a version of Figure 4 with **all** results as supplementary material. Additionally, the criteria for a resulting language to be included needs to be stated more clearly.

Thank you for this important and interesting comment. We would like to emphasize that these emergent languages were only removed in the "closing-the-loop" section (previously Fig. 4, now Fig. 5), and no filtering was performed for the messages generated from the teacher Q-matrices. Therefore, the need to exclude languages when providing the student Q-matrices implies that these languages were specialized to take advantage of the structure of the teacher-provided initial message (see Fig. 1). We found that a large majority of languages (70%) was robust enough to generate meaningful messages even when provided the student information.

Now, let us clarify the exclusion criterion for languages when feeding the student-learned information through the language encoding. Once a set of messages has been generated using the student Q-matrices, we can provide them to the student once more and evaluate their performance on the associated tasks. To then test whether a language was beneficial to the student and allowed for

bi-directional communication, we compared the average performance of the informed student on all the trained tasks and compared this against the average performance of two types of students: 1) a misinformed student (that is provided a similarly distorted message but for an unrelated task), and 2) a random walker. If a language embedding led to a lower performance for the informed student against the higher of the uninformed student or the random walker, we removed this language (see equation below). We included the misinformed student in the criterion, as we assume that if the correct message leads to worse performance than a random message, our language is not beneficial to the student and does not "survive". Additionally, as the reviewer correctly pointed out in a later reply, we included the random walker in this criterion as it is inherently intuitive that the informed student performs better than the misinformed student (who is provided the wrong internal representation) but that the language may nonetheless not provide a competitive advantage over taking random actions. Therefore, we check that (i) the language is inherently functional and (ii) provides information that the student can use. The languages that fail our criterion are those that lead to task information loss or did not imbue the student with generalization abilities (which may arise due to the reasons the reviewer correctly pointed out).

We agree that our initial analogy to natural evolution was not optimal; our intention was to illustrate the principle of selective pressure favoring more adaptive or efficient systems. To address the reviewer's concerns, we have (i) expanded the analogy on natural evolution, (ii) added a supplemental figure of the entire language set (fig. S3), and (iii) attempted to clarify the criteria for exclusion.

The new supplemental figure, which can be seen below on the right next to the relevant panels of Fig. 5 (which only includes the 70% percent of the languages that passed our criterion), shows that in all cases where previously the informed student performed better than the random walker/misinformed student, that relationship is maintained, i.e., even using the sub-optimal language embeddings allows the student to perform better at the tasks. Nonetheless, we also see an expected decrease in the performance of the informed student for all scenarios. This means that we lose that significance for certain situations where previously significantly better performance was observed over the random walker/misinformed student. Additionally, we lessen the generalization capabilities across all but two test tasks (over the smart random walker).

Comparison of student performance when supplied the messages passed through the filtered languages (left, adapted from Fig. 5 in the new manuscript) and when supplied messages passed through all languages (right, new supplemental figure S3).

We added the following text to the updated manuscript:

“These languages were only removed from the set of languages we analyzed in Fig. 5. This language filtering was performed by retaining languages that, on average, led to a higher average task-solving rate for the informed student (receiving the message from encoded student information) compared to the average solving rate of the misinformed student and the random walker (all measured on the trained tasks). We included the misinformed student in the criterion to test whether our language is dysfunctional, i.e., the correct message leads to worse performance than a random message. Additionally, we included the random walker in this criterion as it is inherently intuitive that the informed student should perform better than the misinformed student, but that the language may nonetheless not provide a competitive advantage over taking random actions. Therefore, we check (i) if the language is inherently functional and (ii) if it provides information that the student can use. The languages that fail our criterion are those that lead to task information loss or did not imbue the student with generalization abilities (which may arise due to the reasons the reviewer correctly pointed out). Mathematically, we retain a language if

$$\mathbb{E}[Perf(S_I)] > \max(\mathbb{E}[Perf(S_M)], \mathbb{E}[Perf(\text{Random Walker})])$$

where E is the expected value over all trained tasks, S_I refers to the informed student who is provided the correct distorted message and S_M is the misinformed student. We argue that this can be viewed through the biological lens where selective pressures favor

more adaptive or efficient systems. Akin to the effect of natural evolution, where weak and inefficient members (in this case, languages) die out, languages that are detrimental to the student do not survive. For completeness sake, we have additionally provided a supplemental figure where the full set of languages is displayed in Supplemental Fig. S3. As expected we see a decrease in the performance of the informed student for all scenarios but nonetheless, in all cases where previously the informed student performed better than the random walker/misinformed student, that relationship is maintained, i.e., even using the sub-optimal language embeddings allows the student to perform better at the tasks.”

The manuscript repeatedly states that some higher dimensional meaning space is encoded into a lower dimensional message space, where the lower dimensional message space is supposed to correspond to language. This stands in contrast with the ideas of e.g., word2vec (Mikolov et al., 2013), which encodes the *high-dimensional* language data (think: alphabet size to the power of message length) into a low-dimensional, continuous representation. As the stated goal of the developed framework is to simulate and analyze the emergence of language, the notion/motivation assumed in the manuscript could be described more clearly. In other words, why should the communication channel be low dimensional?

Thank you for mentioning this. To avoid confusion about the number of dimensions present in a language (which may vary by language, context, etc.), we now focus more on animal communication, where prior experimental studies have shown the communication to be low dimensional compared to brain activity. This comparison is critical, as low-dimensional (the communication) is only in reference to a higher-dimensional reference point (brain activity). Nonetheless, this relative comparison does not preclude the lower-dimensional structure from being "high-dimensional" in an absolute number of dimensions. In mammals, communication channels can be the vocal tract or, in general, a limited action space, as we currently state in the discussion of our manuscript:

“This is similar to the natural language, where morphemes evolve according to motives, goals, and efficiency of a group^{30, 62}. For instance, in birds, it has been observed that utility drives the emergence of new linguistic relations or compositions^{63, 64}. Second, introducing dimensionality and sparsity constraints is motivated by anatomical and cognitive limitations, such as vocal tract size or memory capacity^{65, 66}. Hence, by allocating a predefined number of dimensions to our communication system, we replicate such properties and observe that these are organized into hierarchical task-relevant modes.”

We have also included the reviewer’s idea that such a particular number of dimensions within the computational space of a language may be hard to quantify and may be a large number. For example, in the human language, the symbolic space is large. We also want to emphasize that it remains an open question how our symbolic/linguistic space corresponds in terms of dimensionality to neural activity in our brains. For instance, Caucheteux et al., 2023 showed that fMRI dimensionality tends to coincide with the high-dimensional representations shown by machine learning algorithms. As our discussion now reads:

“However, ongoing work still aims to answer how channel size relates to the representational space, as machine-learning and brain activity tend to converge to a high-dimensional space in the representations that are not shared by the actual symbolic space^{50, 67}. Studies have shown that brain activity is compressed relative to the message space even if our languages are not precisely low-dimensional^{10, 48}.”

Thus, we agree with the reviewer that the representational space can have many forms in its dimensionality, as shown by the word2vec community. However, brain representations certainly consider more aspects than the actual morpheme size since it is linked to individual experience and a context that varies with time. In that sense, word2vec seems to show that our linguistic representations tend to overlap in a statistical sense when considered within a linguistic context but our brain activity is composed of multi-regional activations and non-overlapping modes.

Soundness of the methodology

The methodology seems to be for the most part sound. One issue is that the manuscripts state that they use an L2 penalty to promote sparsity. In contrast, it is common to use an L1 penalty to promote sparsity. In contrast, L2 penalty resembles a pressure towards a Gaussian prior, but not necessarily towards sparsity. I suggest to provide more background on why an L2 penalty should promote sparsity in a revised version of this manuscript.

We are grateful to the reviewer for his/her thorough reading of our manuscript. We agree that L1 is the common method to promote sparsity and that we had mentioned L2 in error while referring to L1 (we have tried both during the research phase of the project). We have corrected this error in the manuscript and performed a thorough analysis of the code that generated the results in our paper to ensure that we indeed used the L1 penalty.

Moreover, I'm wondering whether it is reasonable to consider teacher and student as separate agents. The two(?) agents are connected by a continuous channel *and* the gradient is propagated back through this channel. Unless I misunderstood some part of the approach, this would be equivalent to training a single neural network model (with a regularization term on a bottleneck module). If this was the case, it would render the main result little surprising: swapping

out some intermediate representation with an intermediate representation corresponding to a different example (here: task), would certainly degrade the performance.

We thank the reviewer for this insightful comment and for suggesting to clarify the relationship of our setup to a single ANN agent. The approach that the reviewer mentions is related to the DIAL of Foerster et al., 2016, where the gradient information is propagated through all agents. Instead, our approach is closer to RIAL of Foerster et al., 2016 or Tampuu et al. 2017, albeit with a continuous communication interface. Here, the agents each learn their own network parameters, treating the other agents as part of the environment. Below, we discuss why the scenarios we consider cannot be mapped to a single ANN agent, either because the gradient information the reviewer mentions is not passed to the teacher or the messages are frozen, and only their decoding is learned.

Let us discuss the three different approaches we consider: (i) the training of the auto-encoder without the student feedback (all networks are separate) (performance in Figure S4), (ii) connecting the auto-encoder and the student by using the student performance as feedback to the language training (teacher and auto-encoder/student are separate networks) (performance in Figure 4) and (iii) we train the auto-encoder with student feedback; once a certain threshold is reached, we freeze the resulting language and provide that to a new student who has to learn how to interpret this. In this case, the frozen autoencoder and the new student are separate networks without sharing any gradient information (performance in Figure S5).

In each of these three approaches, providing the lower-dimensional message enhances the student's performance, allowing them to outperform the random walkers on a variety of tasks. Furthermore, we note that despite their differences, each approach endows the student with the ability to generalize beyond tasks they have not already seen. In terms of ranking, the best performance (both over the random walkers and in terms of generalization) is achieved by the student who is trained together with the auto-encoder, followed by the student who learns to interpret the frozen language that obtained student feedback, and finally, the student who used the language generated without student feedback.

The fact that the frozen language with student feedback allows the new student to outperform the student who learns to interpret the language without feedback is particularly interesting. This implies that initial feedback helps generate useful language features that are independent of the students.

To clarify, in all three situations, the teacher and the autoencoder/student never share gradient information, making them distinct entities. More concretely, in our three network (teacher, language, student) setup, we have the following key distinctions:

- The teacher agent is always separately trained from the other networks, and the Q-matrices are only passed to the auto-encoder once teacher training is complete. This contrasts a single neural network model where task learning and compression occur simultaneously.
- While the student's performance is considered during the autoencoder training, it is not used to update the teacher. Instead, the student's performance serves as feedback to refine the effectiveness of the message in the autoencoder. This feedback guides the encoding process but does not affect the teacher's ability to perform the task.
- When we link the student performance to enhance the auto-encoder performance, we establish a continuous communication channel between the language and the student that also includes gradient back-propagation. Thus, the auto-encoder includes knowledge from the student network and is structurally equivalent to one single network. We liken this approach to the internal representations of the student being influenced by their actions.

We believe that these contrasting approaches have allowed us to gain insights into the nature of shared representations, including that our representations imbued the students with the ability to generalize and that the student feedback led to a better message encoding (lower reconstruction loss, overall lower SAE loss). This second point is not immediately intuitive, as it is not apparent why the performance of the student should lead to a better representational space. In this case, we argue that the bi-directionality of the communication is key to the success of the language (which in this case can be argued to be internal representations) and student.

We have added a comment to address this topic in the manuscript as follows:

"We note that this framework differs from an approach where all agents and language are connected via one network. Instead, the teachers are always trained separately to generate the relevant task information. Then, we either sequentially train the language and student networks (no feedback in Fig. 1c) or connect the language and the student by providing the auto-encoder feedback on the student performance (Fig. 1c, with feedback). In essence, the teacher and the language (feedback and non-feedback) are connected conceptually through the information transfer process but not in a way that results in a single neural network or shared gradient flow. Variations of this approach were employed by Foerster et al., 2016¹⁵ and Tampuu et al. 2017⁵⁵, which studied "independent Q-learning" where the agents each learn their own network parameters, treating the other agents as part of the environment.

[...]

"We note that the above results arise from a language generated with student feedback, i.e., the representations that help the student have direct knowledge of the student parameters. To ascertain whether this language is useful to students who were not directly involved in the language training, we studied the performance of novel students who were trained to interpret "frozen" languages without feedback (Fig. S4) and with feedback (Fig. S5). We note that the former approach treats all the components of our framework (teacher, language, student) as separate networks and that no gradient information is propagated back through the language channel. We see that in both cases, the students perform well on many tasks and can generalize to unknown scenarios. However, the student who is trained to interpret the frozen feedback language performs better across all scenarios (and outperforms the smart walker). This implies that initial feedback is fundamental to generating helpful language features that other new students can use."

Finally, we updated the first figure, which describes our model setup to clarify the nature of our networks. This involved altering panel c of that figure to demonstrate the separate training phases of each approach.

Figure 1. Teacher-to-student communication model using a continuous compression of task solutions to low-dimensional message vectors. **a)** Model sketch depicting a generalist student agent that is provided messages from teacher agents for various tasks. The student learns to decode these messages and then perform the relevant tasks. **b) Top)** Representative navigation tasks used to train and test agents to analyze the social learning framework. Beginning in the bottom left corner, the agents aim to reach the goal (trophy) in as few steps as possible while avoiding the walls (light blue squares). **Bottom)** Overlaid are example policies for tasks learned as by the teacher agents. The student needs to decode the encoded version of this information it receives. Messages

may contain erroneous instructions or be misunderstood by the student (red squares). **c)** Detailed communication architectures used in this study. In each of the three approaches, task information (Q-matrices in our framework) is first learned by teacher agents who then pass this information through a sparse autoencoder (language proxy), which generates the associated low-dimensional representations, m_i . When student feedback is absent (top row), these representations, m_i , are provided directly to the student who learns to interpret them to solve task i . In the case of student feedback (middle row), we also allow feedback from the student performance to propagate back to the language training and enhance the usefulness of the messages. The final schematic (bottom row) depicts the "closing-the-loop" architecture. Here, the student is trained on a set of messages from expert teacher. Once it is sufficiently competent, its task information is supplied to itself (after being passed through the language embedding, trained with feedback), and the effect on performance is studied.

Reproducibility

The approach and hyperparameters are described in sufficient detail, yet the criteria for hyperparameter selection could be stated more clearly. A sensitivity analysis for the student-feedback weighting factor is provided. However, it could be complemented by analyzing varying other critical hyperparameters, such as the code dimension of the autoencoder. Lastly, for full reproducibility and reuse of the experimental framework, it would be necessary to share the source code.

Thank you very much for these helpful and detailed comments. We agree that an extended hyperparameter study would be valuable and have added this to the supplemental material. This study includes a performance analysis under a variation of three different hyperparameters, which we judged to be most critical to the communication framework. These are the number of training epochs N_{epochs} , the student-feedback weighting factor, ζ , and the dimension of the message, K . We see that the performance results are stable under small variations of the relevant hyperparameters and our selections are all in the region of optimal performance and no overfitting of the training data. We have added this figure to the supplemental section of the updated manuscript.

Figure S8. Effect of varying the key parameters of our communication protocol on the student performance. We tested the effect of varying hyperparameters on the student performance on the trained tasks (a-c, all tasks in 4×4 mazes with 0 or 1 wall state) and on the test tasks (d-f, all tasks in 4×4 mazes with 2 wall states). The three hyperparameters we judged to have the most qualitative impact on the communication protocol were varied, namely the number of training epochs in a), d), the loss weighting ζ in b), e) - see eq. (1) - and the length of the messages in c), f). The hyperparameter values we consistently chose for all our main results are highlighted - see table S2 for reference. We note that in all the majority of the cases, the parameters chosen for our study (black boxes) represent a stable regime.

With regards to the selection criteria, we note that no formal hyperparameter optimization was done, as the purpose of this work was to study the possibilities of our proposed framework, such as the structures of the message space or the effect of student feedback, which we observed were largely independent of the hyperparameters (see figure S6 in the updated manuscript which showed that the student feedback enhanced the auto-encoder loss for a range of ζ values), rather than obtaining optimal performance. We have added this discussion to the supplement as follows:

“The parameters chosen for this work represent a stable regime that allowed us computational tractability while still observing interesting features. Given the stability observed in the hyperparameters for the key features of this work (such as the effect of student feedback on the auto-encoder performance, see figures S6 and S8), we note that changes to these parameters would not have led to significantly different conclusions. Nonetheless, no formal hyperparameter optimization was done, as the purpose of this work was to study the possibilities of our proposed framework, such as the structures of the message space, which we observed were largely independent of the hyperparameters.”

Additionally, following the reviewer’s comment that it would be useful for the full source code to be openly available we have now provided a web link to the repository where the code to train the agents, generate languages and plot the figures is available. This link is now in the “Data and code availability” section of the submitted manuscript. This repository includes a DOI (10.5281/zenodo.7885527), is: https://github.com/meggl23/multi_agent_language/releases/tag/v3.0.

References

Coming back to connecting with the literature: I explicitly do **not** request citing all of this literature but I encourage to look into it and discuss the relationship of the proposed approach to the general themes of a) (discrete) emergent communication and b) (cooperative) multi-agent reinforcement learning, and c) computational modeling in language evolution research. I hope this list is helpful.

- Chaabouni, R., Kharitonov, E., Bouchacourt, D., Dupoux, E., & Baroni, M. (2020). Compositionality and Generalization In Emergent Languages. ACL 2020.
- Chaabouni, R., Kharitonov, E., Dupoux, E., & Baroni, M. (2021). Communicating artificial neural networks develop efficient color-naming systems. Proceedings of the National Academy of Sciences, 118(12).
- Chaabouni, R., Strub, F., Altché, F., Tarassov, E., Tallec, C., Davoodi, E., Mathewson, K. W., Tieleman, O., Lazaridou, A., & Piot, B. (2022). Emergent communication at scale. Proceedings of ICLR.
- Foerster, J., Assael, I. A., de Freitas, N., & Whiteson, S. (2016). Learning to Communicate with Deep Multi-Agent Reinforcement Learning. Advances in Neural Information Processing Systems, 29.
- Galke, L., Ram, Y., & Raviv, L. (2022). Emergent Communication for Understanding Human Language Evolution: What’s Missing? Emergent Communication Workshop at ICLR.
- Gong, T., Shuai, L., & Zhang, M. (2014). Modelling language evolution: Examples and predictions. Physics of Life Reviews, 11(2), 280–302. <https://doi.org/10.1016/j.plrev.2013.11.009>
- Guo, S., Ren, Y., Mathewson, K. W., Kirby, S., Albrecht, S. V., & Smith, K. (2021, October 6). Expressivity of Emergent Languages is a Trade-off between Contextual Complexity and Unpredictability. International Conference on Learning Representations. https://openreview.net/forum?id=WxuE_JWxjkW
- Havrylov, S., & Titov, I. (2017). Emergence of Language with Multi-agent Games: Learning to Communicate with Sequences of Symbols. Advances in Neural Information Processing Systems, 30.
- Kharitonov, E., Chaabouni, R., Bouchacourt, D., & Baroni, M. (2020). Entropy Minimization In Emergent Languages. Proceedings of ICML.
- Kirby, S., & Tamariz, M. (2022). Cumulative cultural evolution, population structure and the origin of combinatoriality in human language. Philosophical Transactions of the Royal Society B: Biological Sciences, 377(1843), 20200319. <https://doi.org/10.1098/rstb.2020.0319>
- Lazaridou, A., Hermann, K. M., Tuyls, K., & Clark, S. (2018). Emergence of Linguistic Communication from Referential Games with Symbolic and Pixel Input. Proceedings of ICLR.
- Li, F., & Bowling, M. (2019). Ease-of-Teaching and Language Structure from Emergent Communication. Advances in Neural Information Processing Systems, 32.
- Lowe, R., Foerster, J., Boureau, Y.-L., Pineau, J., & Dauphin, Y. (2019). On the Pitfalls of Measuring Emergent Communication. AAMAS 2019. <http://arxiv.org/abs/1903.05168>
- Michel, P., Rita, M., Mathewson, K. W., Tieleman, O., & Lazaridou, A. (2023, February 1). Revisiting Populations in multi-agent Communication. Proceedings of ICLR.
- Ndousse, K. K., Eck, D., Levine, S., & Jaques, N. (2021). Emergent Social Learning via Multi-agent Reinforcement Learning. Proceedings of ICML.
- Oroojlooy, A., & Hajinezhad, D. (2022). A review of cooperative multi-agent deep reinforcement learning. Applied Intelligence, 1-46.
- Rita, M., Strub, F., Grill, J.-B., Pietquin, O., & Dupoux, E. (2021, September 29). On the role of population heterogeneity in emergent communication. Proceedings of ICLR.
- Ruis, L., Andreas, J., Baroni, M., Bouchacourt, D., & Lake, B. M. (2020). A benchmark for systematic generalization in grounded language understanding. Advances in Neural Information Processing Systems, 33.
- Tieleman, O., Lazaridou, A., Mourad, S., Blundell, C., & Precup, D. (2019). Shaping representations through communication: Community size effect in artificial learning systems. NeurIPS 2019 workshop on visually grounded interaction and language.
- Tucker, M., Li, H., Agrawal, S., Hughes, D., Sycara, K., Lewis, M., & Shah, J. A. (2021). Emergent Discrete Communication in Semantic Spaces. Advances in Neural Information Processing Systems, 34.

We would like to thank the reviewer for suggesting additional material for our discussion which put our work better into context. We have included most of them in the revised manuscript, especially in the introduction and discussion where we now focus more on Foerster, *et al.*, (2016), Tieleman *et al.*, (2019) and Tucker *et al.*, (2021) while giving credit to other works.

Reviewer #3 (Remarks to the Author):

This paper provides a nice and interpretable framework/model that allows language to emerge in multi-agent systems allowing for the analysis of RL agents that are equipped with the ability to represent and communicate features of an artificial language that directly involves the task they are trained to perform. They show that for tasks represented in teacher/student setting, the student learns to decode messages from the teacher to perform tasks. When measuring % tasks solved, they see that informed students (that get message information) perform significantly better than misinformed students or random walkers, however student-student performance leads to lower performance sometimes. They also analyse the embeddings produced by each model to visualise the differences in representations learned.

I think a lot of the analyses could be framed better to allow a better understanding of the different components of the system. Specific points below:

Some analyses do not seem to be in line with the rest of the results/hypotheses posited here. For example, the PCA plots can allow us to understand if there is a difference in the language embeddings after training and whether embedding cluster in intuitive ways, however the results don't seem very conclusive. It would be good to have a discussion section that elaborates on these, and also a more thorough investigation of different types of dimensionality reduction approaches that might lead to clear results.

We thank the reviewer for his/her thoughtful and constructive feedback on our manuscript. Thank you for the suggestion to dissect the language embedding better, particularly the PCA plots (Fig 2). We note that comparing the language representations arising from the auto-encoder without and with student feedback reveals a fundamental alteration in language representations. Initially, the representations (Fig. 2a) are clustered by the goal location (a direct proxy for the maze task), and within those clusterings, the representations are stratified by the goal location. While this conveys a significant amount of information about the task to the student (similar to a one-hot vector, which provides a label for each world), this does not represent the most useful information to the student. If we contrast this with the PCA projection of the feedback representations in Fig. 2b, we note that the clustering has completely disappeared. Instead, we have a stratification in the goal location (PC 1) and the wall position (PC 2), implying that better student performance is not linked to the wall location but to the actions the student must perform to achieve the goal. This is highlighted by the fourth panel in Fig. 2b, where there is a clear separation in the representations when the first action is up or to the right. Despite the student feedback representations focusing heavily on the possible actions of the student, each individual maze is clearly represented in the lower-dimensional space (see Fig. 2c). Here, the goal location is rotated by 90 degrees, but nonetheless, the wall state can be observed and the square nature of the maze itself. Further evidence of this difference is also demonstrated in Table 1, where the variance between and within clusters is shown.

What we found particularly intriguing was the effect of the lower-dimensional space after being passed through the students in the closing-the-loop analysis. Here, we projected the student's output onto the lower-dimensional PC space and saw that the wall location no longer became essential. Instead, the goal location (first PC) accounted for over 90% of the variance. This further highlighted that in our study, the student's action was much more important than the state in which they found themselves.

We then performed the same analysis using networks with linear and non-linear activation functions to determine whether this effect would persist if we altered our architecture. These results can be seen in the supplemental section (Figs. S1 and S2) of the manuscript and show that similar trends persisted (i.e., a focus on first action in the representations with student feedback).

Finally, we followed the reviewer's suggestion to apply different dimensionality reduction techniques to our results. To this end, we utilized two non-linear approaches (UMAP and T-SNE) and studied the lower dimensional embedding. Both t-SNE and UMAP rely on non-linear optimization to find the lower-dimensional embedding that best represents the input data. These optimization routines depend on tunable parameters that can significantly affect the final embedding space. In the case of UMAP, this parameter is the neighbors parameter, and in the case of t-SNE, it is the perplexity parameter. Both parameters can be regarded as the balance between considering local versus global structure in the data. We chose two different parameters for each method to study the lower-dimensional space. We note that we see that the previously observed characteristics persist:

- Initial clusters in the embeddings are observed across t-SNE and UMAP if no student feedback is provided. These clusters are distinct from each other and can be linked to the one-hot vector approach.
- Within those clusters, high stratification according to goal location is observed.
- Student feedback reduces these clusterings and instead focuses on student action as primary information. This is further emphasized by the clear separation between initially moving upwards or to the right.

- When considering the individual maze representations, we only minimally reconstruct the geometric structure of the maze (in contrast with the representations achieved with the PCA in Fig. 2c, iv). Nonetheless, a clear distinction between initial action is recovered.

We have expanded on these points in the discussion as follows:

“The resulting latent space shows wall positions as the most prominent dimension in the lower-dimensional representations (Fig. 2a(ii)), with goal locations being a secondary feature of the variability (Fig. 2a(iii)). This structure is represented in the lower-dimensional PCA through discrete groupings with minimal overlap and stratification within each grouping according to the goal location. This result follows intuitively from the fact that the language is trained without student feedback, only relying on the reconstruction of the Q-matrix and regularization of the message space (eq. (4)). Thus, to achieve this reconstruction most sparsely, a hierarchical structure appears: first, we distinguish mazes, and then, within each maze, we distinguish the goal location. This structure appears regardless of whether this information is helpful for the student. We note that when we used linear activations or singular value decomposition for the language encoding, we did not reproduce this clear grouping (cf. Fig. S1 and Fig. S2).”

[...]
“Notably, the latent structure of the language space significantly changes through this reward-maximizing term (Fig. 2b(ii)-(iv)). Even if the variance distribution remains similar (compare Fig. 2a(i), Fig. 2b(i)), task settings are no longer clustered in the latent space, but instead form a more continuous gradient when marked by wall position (Fig. 2b(ii)) or goal location (Fig. 2b(iii)). Therefore, the feedback changes the lower-dimensional task representations so that the student obtains more information on where to go, i.e., the policy, rather than the actual composition of the state space. We note some overlap in the middle of the cluster when marking the tasks by goal location; here, the policy differences are negligible as there might be two competing policies that are equally optimal. This focus on policy is additionally emphasized by the variability along the initial action of the student (Fig. 2b(iv)), where a clear split between the two choices of going right or up can be observed. By providing this policy label, language moves away from providing maze labels and towards a framework that can generalize to tasks the student has not seen before. Table 1 shows the changes in explained variability by wall position and goal location in both languages without and with student feedback. Notably, the message variability between groups of goal locations (see Methods) rises when the utility constraint is introduced, marking the increased importance of describing the goal location accurately in the language.

This focus on policy, rather than state space, appears to be independent of the architecture of the autoencoder we use (cf. all linear activations in Fig. S1) or the dimensionality reduction technique we employ (see Fig. S9 - Fig. S12) for results using UMAP and t-SNE). This implies that transmitting this representational feature is fundamental to the success of the student.

“
 Additionally, we added 4 figures that show the analysis using UMAP and T-SNE to the supplemental material, as well as a description of the analysis

“We test two different non-linear dimensionality reduction methods on the messages from a language created without feedback and one created with feedback (the same as in Fig. 2 in the main results). Both methods, t-SNE⁷³ and UMAP⁷⁴, come with tunable parameters, the most crucial of which are the perplexity π (t-SNE) and the number of neighbors k (UMAP), which can be seen as the number of nearest neighbors that are taken into account for each data point when designing the low-dimensional representation of the data. In the following, the subscript “all” on these parameters refers to the case of messages from all trained worlds, whereas the subscript “single” refers to the case of messages from only a single world. We experiment with comparatively (with respect to the number of samples) low (Fig. S9, Fig. S11) and comparatively high (Fig. S10, Fig. S12) values for π and k , respectively. All other parameters, namely the number of iterations in t-SNE (= 1500) and the minimum distance between data points in UMAP (= 0.5), are kept constant.”

Figure S9. t-SNE of messages, low perplexity parameter. We apply the t-SNE dimensionality reduction method (perplexities $\pi_{all} = 10$ and $\pi_{single} = 2$) to both the language created without student feedback and the one created with student feedback from Fig. 2 in the main results. The individual panels are conceptually identical to Fig. 2: **a)** shows all messages from the language created without feedback, **b)** all messages from the language created with feedback and **c)** those messages from the language analyzed in b), which refer to tasks in a single maze shown in c)i. The coloration of the messages is done by wall position, goal location of the task and (in the cases of feedback) the probability of the first student action, respectively.

Figure S10. t-SNE of messages, high perplexity parameter. We apply the t-SNE dimensionality reduction method (perplexities $\pi_{all} = 50$ and $\pi_{single} = 5$) to both the language created without student feedback and the one created with student feedback from Fig. 2 in the main results. The individual panels are conceptually identical to Fig. 2: **a)** shows all messages from the language created without feedback, **b)** all messages from the language created with feedback and **c)** those messages from the language analyzed in b), which refer to tasks in a single maze shown in c)i. The coloration of the messages is done by wall position, goal location of the task and (in the cases of feedback) the probability of the first student action, respectively.

UMAP of messages with no student feedback

UMAP of messages with student feedback

Figure S11. **UMAP of messages, low neighbors parameter.** We apply the UMAP dimensionality reduction method ($neighbors\ k_{all} = 10$ and $k_{single} = 2$) to both the language created without student feedback and the one created with student feedback from Fig. 2 in the main results. The individual panels are conceptually identical to Fig. 2: **a)** shows all messages from the language created without feedback, **b)** all messages from the language created with feedback and **c)** those messages from the language analyzed in **b)**, which refer to tasks in the maze shown in **c) i**. The coloration of the messages is done by wall position, goal location of the task and (in the cases of feedback) the probability of the first student action, respectively.

UMAP of messages with no student feedback

UMAP of messages with student feedback

Figure S12. **UMAP of messages, high neighbors parameter.** We apply the UMAP dimensionality reduction method ($neighbors\ k_{all} = 50$ and $k_{single} = 5$) to both the language created without student feedback and the one created with student feedback from Fig. 2 in the main results. The individual panels are conceptually identical to Fig. 2: **a)** shows all messages from the language created without feedback, **b)** all messages from the language created with feedback and **c)** those messages from the language analyzed in

b), which refer to tasks in the maze shown in c) i. The coloration of the messages is done by wall position, goal location of the task and (in the cases of feedback) the probability of the first student action, respectively

There is also the question of why the autoencoder architecture is important to this setup—most other papers try to show results that are model/architecture-agnostic (e.g., this multi-agent communication setting should work with an RNN/LSTM/Transformer or any model with RL feedback) so it would be good to see that these hold up for other architectures and we can make the same claims about language/communication.

The reviewer correctly mentions that other models tend to be architecture-agnostic when referring to the channel type. This is because either they opt for some symbolic representation or they choose a more “standard” network system for the transmission – e.g., Foerster *et al.* (2016), Tieleman *et al.* (2019), and Tucker *et al.* (2021). In our case, we are interested in analyzing the representational embedding inside an agent, its subsequent compression into a message, and its action on the neural space of the student. We chose an autoencoder-mediated setting because it permits us to constrict the message space while also operating in an ‘interpretable’ linear regime that yields an easy comparison or relation between the internal and state-space domains.

In future studies, one can generalize our approach to sequential systems, as proposed by the reviewer, that produce latent spaces that are dynamical and compositional. However, the presence of internal dynamics and connectivity initialization makes the interpretability more complex compared to the simple feedforward autoencoder, where the dimensional bottleneck is by design and not by training.

Line by line comments:

1. Page 1: “Language in a broad sense (citation 9)” → it is unclear what this means given the citation is about the use of language in grounded situations and with a communicative intent, rather than what this sentence implies it is being cited for. It might be worth rephrasing the sentence here to correctly reference the cite mentioned.

We thank the reviewer for spotting this. We agree that the original sentence, in combination with the citation, needed to be clarified. The citation refers to situations with a communicative intent but also discusses the concept of language in the broad sense and in the narrow sense, where “narrow sense” refers to the grammatical aspects of language, such as syntax, while “broad sense” includes a wider range of language-related abilities such as representational embeddings.

In our work, we thus aimed to study language in this broader sense as a cognitive faculty encompassing more than just communicative elements. Therefore, this “language”, which is not focused on a fixed set of concrete symbols, includes concepts such as a low-dimensional embedding space of messages that allows for shared representations of tasks or objects among agents.

We have amended this phrase in the introduction as follows:

“Here, we focus on studying an emergent language in a broad sense⁹, which refers to the ample brain-based linguistic abilities shared across animal species. Thus, rather than modeling concrete symbols or grammar, we assume language is a common low-dimensional embedding facilitating the exchange of representations across individuals. Using this definition, we can then study the co-evolution of task abstractions among different agents and how sharing those representations affects agent performance.”

2. “Surge of papers that attempt to combine linguistic properties with deep neural network...” → a better rephrasing is “papers that combine multi-agent systems with communication policies.” The current sentence is slightly incorrect since the current phrasing (linguistic properties + deep learning) is more applicable to the entire field of NLP rather than just the emergent communication multi-agent work that the authors should be citing here

We agree, our previous wording might have been too general. We have now implemented this suggestion in the updated version of the manuscript.

3. The last paragraph before “model architecture” can be written better to concretely summarise the goals. Maybe a bullet point list of the main contributions of this paper would be helpful to make this paper clearer.

Thank you for this suggestion, we have implemented this as suggested by providing a bullet point list of the main contributions of our work as follows:

“To summarize, we present a tractable framework for understanding emergent communication, disentangling the relation between the internal neural representations and the message space that contributes following three novel results to the neuroscience and neuroAI communities:

- *reveal features in the lower-dimensional embedding space necessary for higher student success and task generalization^{6,43}.*
- *demonstrate that the communication is enhanced and the information content of the message increased when the communication channel is provided with feedback in the form of the student’s task performance.*

- provide an avenue to studying symmetric communication, i.e., where the sender and receiver can be interchanged, highlighted recently as an important challenge in the field³³.

Let us note that for readability and to meet journal style requirements, it is possible that we will be asked to remove the bullets at the beginning of each line, but this will not change the content of the paragraph and its bullet-style nature.

4. “Natural language is a lower dimensional representation of higher level concepts” → cite to Manning et. al., does not seem very well founded. The textbook cited gives an overview of statistical NLP/parsing/other algorithms, but is not a good source of the sentence written here. It is not a widely held hypothesis that natural language is “lower dimensional” than any concepts, so if the authors want to claim this it would be worth arguing this with the correct citations.

We thank the reviewer for suggesting to replace Manning et al. with a better citation. Hence, we have substituted it with citations measuring the entropy reduction or dimensionality of language (in some instances in parallel with brain recordings) more directly. The current manuscript reads:

“When one individual speaks to another, high-dimensional descriptors – e.g., time, location, shape, context – of a concept in the brain of the sender are encoded into a low-dimensional vocabulary that is decoded back into a higher-dimensional and distributed representation in the brain of the receiver. This is supported by the observed semantic correlations and low-dimensional embedding space of human language representations^{40, 47, 48} and in the brain activity^{10, 49, 50}, which is congruent across species^{6, 51}.”

40. Bengio, Y., Ducharme, R. & Vincent, P. A neural probabilistic language model. *Adv. neural information processing systems* 13 (2000).

47. Rocktäschel, T., Bošnjak, M., Singh, S. & Riedel, S. Low-dimensional embeddings of logic. *Annu. Meet. Assoc. for Comput. Linguist.* DOI: 10.3115/v1/w14-2409 (2014).

48. Antonello, R., Turek, J., Vo, V. A. & Huth, A. G. Low-dimensional structure in the space of language representations is reflected in brain responses. *Neural Inf. Process. Syst.* (2021).

10. McKenzie, S. et al. Hippocampal representation of related and opposing memories develop within distinct, hierarchically organized neural schemas. *Neuron* DOI: 10.1016/j.neuron.2014.05.019 (2014).

49. Huth, A. G., De Heer, W. A., Griffiths, T. L., Theunissen, F. E. & Gallant, J. L. Natural speech reveals the semantic maps that tile human cerebral cortex. *Nature* 532, 453–458 (2016).

50. Caucheteux, C. & King, J.-R. Brains and algorithms partially converge in natural language processing. *Commun. biology* 5, 134 (2022).

6. Hasson, U., Ghazanfar, A. A., Galantucci, B., Garrod, S. & Keysers, C. Brain-to-brain coupling: a mechanism for creating and sharing a social world. *Trends cognitive sciences* 16, 114–121 (2012).

51. Robotka, H. et al. Sparse ensemble neural code for a complete vocal repertoire. *Cell Reports* 42 (2023).

5. “Ought to be generalisable” → it would be nice to frame this by drawing an analogy to humans or our expectation of why we would want task generalisation.

We agree with the reviewer that the addition of a human/artificial agent analogy would better underline the importance of generalization. We now write:

“Humans take advantage of this generalizability to perform new or slightly different tasks from the ones they may have encountered before. For instance, when learning to ride a bicycle, an individual does not need to relearn all the principles of balance and coordination when switching to a different bike or even another mode of transportation, like a scooter or a motorcycle. Similarly, an artificial agent faced with an out-of-distribution task may need to draw on its internal representations and their generalizability to complete it successfully.”

REVIEWER COMMENTS

Reviewer #1 (Remarks to the Author):

I have read the authors' response and the revised manuscript. All my initial concerns have been addressed in the revision. I have only one more minor point regarding the text added in the revision:

Reference [8]: While I understand the authors intent to clarify the terminology of 'language in a broad sense', the referenced paper emphasizes that the Faculty of Language in a Broad Sense, which "includes a sensory-motor system, a conceptual-intentional system, and the computational mechanisms for recursion, providing the capacity to generate an infinite range of expressions from a finite set of elements." [from the abstract]. I'm not sure if this is a good fit for what it is cited for in the submitted paper. If the authors agree, I suggest to consult (or, contextualize with) Hockett, C. The Origin of Speech (1960) and/or some more recent perspectives such as Dupoux, E. (2018). Cognitive Science in the era of Artificial Intelligence: A roadmap for reverse-engineering the infant language-learner. *Cognition*, 173, 43–59. <https://doi.org/10.1016/j.cognition.2017.11.008>

Reviewer #3 (Remarks to the Author):

This paper focuses on a cooperative task to allow two agents to train with reinforcement learning to develop a communication protocol with each other, and shows how the structure of the message/communication space can be important to better task performance.

Reasons to accept:

1. This paper focuses on an important and well-studied task in the multi-agent communication literature i.e., training teacher-student agent networks to develop a communication protocol to cooperate with each other to solve a task.
2. The task/experimental set up is straightforward and simple enough that it allows a clear analysis of different components of the system
3. The models used (SAEs) and RL algorithms (Q-learning) are also well established and used in the literature previously (although other networks also have been explored by several other works)
4. The results are intuitive and in-line with previous work—the authors show that student feedback incorporated into teacher training increase the success rate, but additionally their novel contribution is that they can make claims about how the structure of the latent space is changed by this training protocol and the presence/absence of messages

Limitations:

1. This paper misses some important citations that also use natural language/multi-agent settings that have relevant insights. See e.g., the paper here <https://aclanthology.org/2020.acl-main.685.pdf> and a more detailed survey paper here outlining all works on natural language communication/pragmatics/multi-agent <https://arxiv.org/abs/2211.08371> both of which should be cited

2. A more thorough evaluation of different types of models could be explored to help understand the structure of the latent space better. E.g., any neural network with hidden representations that are of a higher dimension than the input discrete language could be subject to an analysis of these representations, and it would be good to see that the results hold up over all of these
3. A deeper understanding of what these representations hold (apart from just PCA) would be helpful to see. (also see e.g., this paper on the limitations of pca analyses <https://arxiv.org/abs/2312.03656>)
4. Approaches such as linear probing classifiers over the representations (to see what information can be decoded/are containing in the reps) would be helpful to glean further insight.
5. The current abstract/key contributions of this paper are hard to discern from previous work. E.g., the authors claim that a big contribution of this work is the introduction/proposal of a framework that allows teacher-student agents to train together to develop a communication protocol (and make use of natural language/generation models) however this is not a novel contribution of this paper, and has been done several times in the literature. This is not to say this paper doesn't have novel contributions that add to the field (it does!) but it would be good to very clearly state those, and also draw from what has already been done, to make the new aspects of this paper extremely clear to any reader. I urge the authors to rephrase the abstract and introduction by 1) clarifying what parts of this framework have been introduced before (and what those limitations are) 2) clearly stating what the addition of this paper is.

Dear reviewers,

Thank you for the many helpful comments and the opportunity to resubmit a revised version of our manuscript.

Below, we address all comments of the reviewers point-by-point. The comments of the reviewers are displayed below in **bold**, our reply in regular font and the corresponding manuscript changes in *italic*.

Reviewer #1 (Remarks to the Author):

I have read the authors' response and the revised manuscript. All my initial concerns have been addressed in the revision. I have only one more minor point regarding the text added in the revision:

Reference [8]: While I understand the authors intent to clarify the terminology of 'language in a broad sense', the referenced paper emphasizes that the Faculty of Language in a Broad Sense, which "includes a sensory-motor system, a conceptual-intentional system, and the computational mechanisms for recursion, providing the capacity to generate an infinite range of expressions from a finite set of elements." [from the abstract]. I'm not sure if this is a good fit for what it is cited for in the submitted paper. If the authors agree, I suggest to consult (or, contextualize with) Hockett, C. The Origin of Speech (1960) and/or some more recent perspectives such as Dupoux, E. (2018). Cognitive Science in the era of Artificial Intelligence: A roadmap for reverse-engineering the infant language-learner. Cognition, 173, 43–59. <https://doi.org/10.1016/j.cognition.2017.11.008>

We thank the reviewer for mentioning that our revision was successful in addressing all previous comments. We followed the recommendation replacing the discussion with regard to "Language in a Broad Sense" to better match the intention of our manuscript. Following the specific suggestions, we now included a citation to "Hockett, C. The Origin of Speech (1960)" and contextualized it in the text. We have also removed the sentence regarding "language in a broad sense" from the introduction and instead included a discussion of the features mentioned by Hockett as being critical for speech-based communication, which are the features our framework replicates. The new text is now as follows:

"To understand the interplay between the environmental experiences and the internal abstractions, we build on a teacher-student framework to develop a communication protocol that allows agents to cooperate while solving a common task^{27,28}. We employ a reinforcement learning (RL) framework²⁹, which has been previously used in artificial agents^{21,30}, to produce empirical task abstractions that vary among agents. Using this RL-based student-teacher framework, we can recapitulate features considered to be critical for language³¹, including "interchangeability"³², where individuals can both send and receive messages³³, "total feedback," where speakers can hear and internally monitor their own speech³⁴ or "productivity," where individuals can create and understand an infinite number of messages that have not been expressed before³⁵."

where citation 31 is the work of Hockett. We have also added references to these concepts (e.g. total feedback) to the discussion of the results in the updated manuscript. Additionally, we are grateful to the reviewer for making us aware of Dupoux's work, which complements our results and goals, and which we now have cited. We have added a sentence to the discussion that considers possible additional future explorations of Dupoux's work using our proposed framework:

"Finally, as introduced by Dupoux⁷⁸, there are several features critical for the study of language emergence and language learning: (i) being computationally tractable, (ii) using realistic tasks that can be performed by real biological agents and (iii) use the results of biological agents as benchmarks for the artificial agent performance. In this sense, while tractability is a key component of our framework, we emphasize its utility to neuroethologists, who can work with biological data within our framework to study brain activity in relation to language abilities in future studies."

Reviewer #3 (Remarks to the Author):

This paper focuses on a cooperative task to allow two agents to train with reinforcement learning to develop a communication protocol with each other, and shows how the structure of the message/communication space can be important to better task performance.

Reasons to accept:

- 1. This paper focuses on an important and well-studied task in the multi-agent communication literature i.e., training teacher-student agent networks to develop a communication protocol to cooperate with each other to solve a task.**
- 2. The task/experimental set up is straightforward and simple enough that it allows a clear analysis of different components of the system**
- 3. The models used (SAEs) and RL algorithms (Q-learning) are also well established and used in the literature previously (although other networks also have been explored by several other works)**
- 4. The results are intuitive and in-line with previous work—the authors show that student feedback incorporated into teacher training increase the success rate, but additionally their novel contribution is that they can make claims about how the structure of the latent space is changed by this training protocol and the presence/absence of messages**

Limitations:

1. This paper misses some important citations that also use natural language/multi-agent settings that have relevant insights. See e.g., the paper here <https://aclanthology.org/2020.acl-main.685.pdf> and a more detailed survey paper here outlining all works on natural language communication/pragmatics/multi-agent <https://arxiv.org/abs/2211.08371> both of which should be cited

We are grateful to the reviewer for making us aware of these studies which we have added to the introduction, where we discuss the current literature. A new sentence in the introduction reads:

“Further work has highlighted the importance of pragmatic approaches²⁴, the contrast between scientific or applied models in language emergence²⁵, and multi-agent cooperative learning²⁶.”

Inspired by the reviewer’s comment we added three more recent, emerging studies to better ground our work in the current literature:

26. Haber, J. et al. *The photobook dataset: Building common ground through visually-grounded dialogue*. arXiv preprint arXiv:1906.01530 (2019).

27. Ku, A., Anderson, P., Patel, R., Ie, E. & Baldridge, J. *Room-across-room: Multilingual vision-and-language navigation with dense spatiotemporal grounding*. arXiv preprint arXiv:2010.07954 (2020).

28. Andreas, J. *Language models as agent models*. arXiv preprint arXiv:2212.01681 (2022).

Finally, as suggested by the first reviewer, we added a reference to the work of Dupoux, E. (2018). “*Cognitive Science in the era of Artificial Intelligence: A roadmap for reverse-engineering the infant language-learner*” which proposes certain features that should be included in any computational approach to language learning/modeling. To help future work build on the framework we have introduced here, we added a phrase discussing possible generalizations as follows:

“Finally, as introduced by Dupoux²⁸, there are several features that are critical for the study of language emergence and language learning: (i) being computationally tractable, (ii) using realistic tasks that can be performed by real biological agents and (iii) use the results of those biological agents as benchmarks for the artificial agent performance. In this sense, while tractability is a key component of our framework, we emphasize its utility to neuroethologists, who can work with biological data within our framework to study brain activity in relation to language abilities in future studies.”

2. A more thorough evaluation of different types of models could be explored to help understand the structure of the latent space better. E.g., any neural network with hidden representations that are of a higher dimension than the input discrete language could be subject to an analysis of these representations, and it would be good to see that the results hold up over all of these

We are grateful to the reviewer for this suggestion to enhance our work. We agree that testing our framework across other network configurations, which will also contain hidden representations, and observing the results would be valuable to validate the robustness of our conclusions. To this end, we repeated the analysis with three new student network configurations. As we had previously altered the size K of the middle hidden layer in the auto-encoder (AE) (see supplemental Figure 8f) and observed that the student performance increased marginally as a wider layer was used in the AE, we decided instead to study the effect of altering the student architecture and observe how the altered feedback would change the AE representations. In this case, we use our “with feedback” approach (see Figure 1c). Thus, the teacher networks and all other hyperparameters were left unaltered. We can now compare them with the results of the original architecture (which we refer to as “regular width” ANNs which has three hidden layers with 10, 20, and 20 neurons, respectively - see Fig 2b and c for results).

The three new student architectures we use to study both the effect of width and depth on the student network are:

- Half width: all hidden layers of the student are half as wide (i.e., three hidden layers of 5, 10, and 10 neurons, respectively),
- Double width: all hidden layers of the student are double as wide (i.e., three hidden layers of 20, 40, and 40 neurons, respectively),
- Single layer: there is only one hidden layer of neurons in the student network (10 neurons)

The result of this analysis using these new network architectures can be seen in the figures below. We begin by focusing on the structure of the latent message space and generating figures comparable to the results presented in Fig. 2b,c (we have added this figure below and titled it “regular width” for comparison purposes).

We note that the main features of the message space structure, namely the focus on policy instead of map layout and the strong influence of the student's first action, are preserved across the different architectures. Additionally, when considering one particular maze, the nature of the two-dimensional representation of the state space remains stable. Interestingly, the least and most interpretable message spaces arise from the single-layer and double-width student, respectively. This implies that the network architecture and the quality of the representations rely on the information-carrying capabilities of the network.

regular width

single layer

half width

double width

To understand next how these different representational spaces would affect student performance and generalizability of the language, we trained a set of different languages (5) per architecture and studied the student's performance on the training and test tasks. Illustratively, we only analyzed the students trained on all goal locations. As shown below, the train and test performances are

almost equal across architectures; only a slight dip can be observed in the "single layer" setting. This implies that while the representational space is degraded for the smaller architectures, it still contains sufficient information to help the student perform their task. We find that this result fits well into our current supplemental figure regarding our hyperparameter tuning (Fig S8), so we have included it in this figure.

✖ informed student ✖ misinformed student value in main results
✖ smart random walker ✖ random walker

student performance

As a final step, we repeated the analysis in the "closing the loop" setting (see Fig. 5). Here, the student networks are given their own representations, which have been encoded using the trained autoencoder. Again, we note that while information content is significantly degraded, the message retains pertinent task information. This task information means that the students who receive the correct messages perform better than random or misinformed students. However, the informed student is only able to outperform the smart random walker (who avoids walls) in the "training" tasks. We emphasize that in this setting, the trained tasks refer to the student being trained on the optimal messages that are the encoded teacher representations.

✖ informed student ✖ misinformed student value in main results
✖ smart random walker ✖ random walker

closing the loop - filtered languages

The above closing-the-loop procedure required us to remove some detrimental languages (for the criterion of removing such a language in the Language filtering section in the Methods). To see whether the structures of the different lower-dimensional message spaces meant that more or fewer languages needed to be filtered out, we calculated the percentage of dropped languages per architecture and plotted this below. Indeed, we see that the "single layer" and "half width" architectures led to a substantial increase in detrimental languages, implying that a certain minimal network width and depth is required for sufficient generalizability of the languages to succeed in closing the loop between teacher and student.

closing the loop - dropout percentage

Given the analysis here, we identified a minimum depth necessary to maintain sufficient task information in the lower-dimensional message space. Nonetheless, even in the single-layer student architecture, the message provided useful information and allowed the student to perform well on the test tasks. Given these results, we are confident that (when altering the width and depth of the architecture), our framework would perform in a manner consistent with the results presented in our manuscript.

We agree with the reviewer’s assessment that showing that our framework is model-agnostic (e.g. does not rely on a specific architecture or unique width and depth of agent) and have thus added brief reference to the possible neural architectures for future directions:

“Additionally, it would be interesting to examine more model-agnostic outcomes using sequential network architectures, e.g., recurrent neural networks or transformers”

3. A deeper understanding of what these representations hold (apart from just PCA) would be helpful to see. (also see e.g., this paper on the limitations of pca analyses <https://arxiv.org/abs/2312.03656>)

Thank you for mentioning that PCA can present some limitations in studying the representational space. Thus, we included t-SNE and UMAP as non-linear alternatives to PCA in the supplement of the revised manuscript (see the section "Dimensionality reduction methods"). The opening sentence of this section now reads:

"To gain a deeper understanding of the hidden representations (see, for example, the limitations of PCA analysis⁷⁹) we perform linear discriminant analysis (LDA) as well as two different non-linear dimensionality reduction methods on the messages from a language created without feedback and one created with feedback (the same as in Fig. 2 in the main results)."

Where 79 is the reference provided by the reviewer. Applying all these different methods, we found that these approaches yielded the same qualitative characteristics as PCA. We include this result in the main text:

"This focus on policy, rather than state space, appears to be independent of the architecture of the autoencoder we use (cf. all linear activations in Fig. S1) or the dimensionality reduction technique we employ (see Fig. S9 - Fig. S12) for results using UMAP and t-SNE). Additionally, the projection of messages to the main dimensions of a linear decoder was consistent with the unsupervised representational space (Fig S13). This implies that transmitting this representational feature is fundamental to the success of the student."

Following the reviewer’s suggestion below, we apply linear discriminant analysis (LDA) to provide a new interpretable subspace and how it relates back to PCA.

4. Approaches such as linear probing classifiers over the representations (to see what information can be decoded/are containing in the reps) would be helpful to glean further insight.

We would like to thank the reviewer for both suggestions on analyzing the representational space. We have now incorporated a two-dimensional linear discriminant analysis (LDA) decoding of the message space in Figure S13. Here, we present the encoding shown in Figure 2 but projected to an LDA subspace.

The LDA approach reveals a similar type of encoding to PCA. In particular, we find that, in the entire dataset, the wall position has perfect accuracy in the case of no student and 99.6% for the one with student feedback. Goal location presents a lower accuracy in the absence of a student, concretely, 28.9% versus 75.6% across the entire messages set.

Taken together, these results point to an enriched representational space of the task in terms of the variables decoded. These representations in the presence of student feedback, when the subspace reconstruction follows a reward/utility function, tracks or decodes more variables of the task itself that are productive to agents.

5. The current abstract/key contributions of this paper are hard to discern from previous work. E.g., the authors claim that a big contribution of this work is the introduction/proposal of a framework that allows teacher-student agents to train together to develop a communication protocol (and make use of natural language/generation models) however this is not a novel contribution of this paper, and has been done several times in the literature. This is not to say this paper doesn't have novel contributions that add to the field (it does!) but it would be good to very clearly state those, and also draw from what has already been done, to make the new aspects of this paper extremely clear to any reader. I urge the authors to rephrase the abstract and introduction by 1) clarifying what parts of this framework have been introduced before (and what those limitations are) 2) clearly stating what the addition of this paper is.

We thank the reviewer for their invitation to clarify the strengths and limitations of our work. We agree that our previous abstract and introduction may not have been clear enough with respect to the previous implementations/results of the literature and how we differentiated from that work. Therefore, we have adapted the abstract to highlight this novelty as follows:

Neural systems have evolved not only to solve environmental challenges through internal representations but also, under social constraints, to communicate these to conspecifics. In this work, we aim to understand the structure of these internal representations and how they may be optimized to transmit pertinent information from one individual to another. Thus, we build on previous teacher-student communication protocols to analyze the formation of individual and shared abstractions and their impact on task performance. We use reinforcement learning in grid-world mazes where a teacher network passes a message to a student to improve task performance. This framework allows us to relate environmental variables with individual and shared representations. We compress high-dimensional task information within a low-dimensional representational space to mimic natural language features. In coherence with previous results, we find that providing teacher information to the student leads to a higher task-completion rate and an ability to generalize tasks it has not seen before. Further, optimizing message content to maximize student reward improves information encoding, suggesting that an accurate representation in the space of messages requires bi-directional input. These results highlight the role of language as a common representation among agents and its implications on generalization capabilities.

Additionally, we have altered the relevant paragraphs of the introduction to focus on what was previously found in the literature, how we take advantage of these well-established results and then finally the novelty of our work. We have adapted the final bullet points of the reviewer to focus on the analysis of the latent space structure:

To understand the interplay between the environmental experiences and the internal abstractions, we build on a teacher-student framework to develop a communication protocol that allows agents to cooperate while solving a common task^{27,28}. We employ a reinforcement learning (RL) framework²⁹, which has been previously used in artificial agents^{21,30}, to produce empirical task abstractions that vary among agents. Using this RL-based student-teacher framework, we can recapitulate features considered to be critical for language³¹, including "interchangeability"³², where individuals can both send and receive messages³³, "total feedback," where speakers can hear and internally monitor their own speech³⁴ or "productivity," where individuals can create and understand an infinite number of messages that have not been expressed before³⁵.

In contrast with previous work, we focus on understanding how hidden representations can be shared between agents and what effect the structure of the lower-dimensional language space is. We are particularly interested in three aspects: how agents internally abstract real-world variables, how these abstractions translate into a common, shareable language, and the interaction of these elements. Hence, we opted for a non-discrete language model to directly compare the continuous nature of both brain processes and real-world phenomena. By feeding into the language model the learned information provided by RL agents performing a navigational task³⁶, we investigated the development of natural language as it arises from social and decision-making processes³⁷. This leads to individualized abstractions emerging organically rather than being pre-defined, in contrast to supervised learning methods^{22,38,39}. By analyzing the structure of the language embedding, we can gain insights into information content in the message space and its relation to neural representations underpinning task performance and generalization. Additionally, it stands apart from previous non-supervised, symbolic methods^{28,40-43}, taking cues from continuous language generation models^{44, 45} and animal communication systems, like the bee waggle dance, which translate a continuous environment into a concise message space^{46,47}, also seen in human languages^{44,45}.

To summarize, we present a tractable framework for studying emergent communication, drawing upon an established multi-agent language model. We disentangle the relation between the internal neural representations and the message space, contributing the following three novel results to the neuroscience and neuroAI communities:

- reveal structural features in the lower-dimensional embedding space necessary for higher student success and task generalization^{6,48}.*
- demonstrate how the structure of the lower-dimensional embedding or message space is altered to enhance the information content when the communication channel is provided with feedback to optimize student performance*
- understand how the hidden representations can be used in studying symmetric communication, i.e., where the sender and receiver can be interchanged, highlighted recently as an important challenge in the field³².*

REVIEWERS' COMMENTS

Reviewer #1 (Remarks to the Author):

The authors have addressed all points raised by previous reviews.

Specifically, the authors have contextualized their work with the suggested literature (and more). In addition, the authors have added a comprehensive study of different model configurations (regular width, single layer, half width, and double width). The study of other model architectures is pointed out as future work, which is fair. Moreover, the authors have consulted different techniques to analyze the latent space (t-SNE, UMAP) to complement their PCA analysis as requested – and leading to the same conclusions. The authors have added a Linear Discriminant Analysis in response to the request for linear probes, which is a reasonable choice to implement linear probes in their setting – again supporting the paper's findings. Lastly, the authors have responded to the request to change the abstract and have also edited parts of the introduction to ensure that the paper's framing accurately represents the results.

All in all, I believe that all comments are addressed and I have no further comments to raise, except a small request to include a requirements.txt file in the GitHub repository for reproducibility.

Reviewer #1 (Remarks on code availability):

The code is conveniently arranged in a single Jupyter notebook file. The code is clearly organized and well-documented. Instructions for producing Figures of the paper are provided. For ease of re-use, I recommend to add a requirements.txt file that holds the dependencies including their version numbers.

Dear reviewer,

Thank you for the many helpful comments and the opportunity to resubmit a revised version of our manuscript.

Below, we address all comments of the reviewers point-by-point. The comments of the reviewers are displayed below in **bold**, our reply in regular font and the corresponding manuscript changes in *italic*.

Reviewer #1 (Remarks to the Author):

The authors have addressed all points raised by previous reviews.

Specifically, the authors have contextualized their work with the suggested literature (and more). In addition, the authors have added a comprehensive study of different model configurations (regular width, single layer, half width, and double width). The study of other model architectures is pointed out as future work, which is fair. Moreover, the authors have consulted different techniques to analyze the latent space (t-SNE, UMAP) to complement their PCA analysis as requested – and leading to the same conclusions. The authors have added a Linear Discriminant Analysis in response to the request for linear probes, which is a reasonable choice to implement linear probes in their setting – again supporting the paper's findings. Lastly, the authors have responded to the request to change the abstract and have also edited parts of the introduction to ensure that the paper's framing accurately represents the results.

All in all, I believe that all comments are addressed and I have no further comments to raise, except a small request to include a requirements.txt file in the GitHub repository for reproducibility.

Reviewer #1 (Remarks on code availability):

The code is conveniently arranged in a single Jupyter notebook file. The code is clearly organized and well-documented. Instructions for producing Figures of the paper are provided. For ease of re-use, I recommend to add a requirements.txt file that holds the dependencies including their version numbers.

We thank the reviewer for their kind words and are pleased that our changes have sufficiently addressed their concerns. Our work has been significantly improved by their suggestions.

We agree that a requirements file in the GitHub repository would increase reproducibility. To this end, we have added this file as well as included the requirements in the README of the repository.